# The extreme yet transient nature of glacial erosion

H. Patton [1] ✉, A. Hubbard[1,2], J. Heyman[3], N. Alexandropoulou[1], A. P. E. Lasabuda [4,5], A. P. Stroeven [6,7], A. M. Hall [6,8], M. Winsborrow[1], D. E. Sugden[8], J. Kleman[6] & K. Andreassen[1]

Ice can sculpt extraordinary landscapes, yet the efficacy of, and controls governing, glacial erosion on geological timescales remain poorly understood and contended, particularly across Polar continental shields. Here, we assimilate geophysical data with modelling of the Eurasian Ice Sheet − the third largest Quaternary ice mass that spanned 49°N to 82°N − to decipher its erosional footprint during the entire last ~100 ka glacial cycle. Our results demonstrate extreme spatial and temporal heterogeneity in subglacial erosion, with rates ranging from 0 to 5 mm a$^{-1}$ and a net volume equating to ~130,000 km$^3$ of bedrock excavated to depths of ~190 m. A hierarchy of environmental controls ostensibly underpins this complex signature: lithology, topography and climate, though it is basal thermodynamics that ultimately regulates erosion, which can be variously protective, pervasive, or, highly selective. Our analysis highlights the remarkable yet fickle nature of glacial erosion − critically modulated by transient ice-sheet dynamics − with its capacity to impart a profound but piecemeal geological legacy across mid- and high latitudes.

As pervasive geological agents of erosion and deposition, ice sheets can drive the evolution of continental shields, shelves, mountain ranges and passive margins. However, the long-term erosive efficacy of ice sheets has been questioned, particularly across low-relief continental shields and high latitudes with cold subglacial regimes[1–3]. Moreover, despite half a century of thorough investigation, the drivers of, and controls on, glacial erosion are still unclear[4], with − to differing degrees − assertion of the primary dominance of tectonics, lithology, climate and other proxies over ice-sheet dynamics. For example, it has been argued that rates of glacial erosion that exceed 1 mm a$^{-1}$ are driven by rapid tectonic uplift, focused on active margins in maritime/temperate climatic settings where fast flow is promoted by high mass-balance turnover and temperate basal regimes that are independent of subglacial processes per se[5–9]. Likewise, a glacial-buzzsaw mechanism

has also gained traction, where it is argued that climate is the critical control restricting the maximum elevations that mountain regions can attain. Here, mountain glaciers and ice caps − self-limited by their equilibrium line altitude − implicitly designate a climatic/altitudinal threshold at and above which they are highly effective erosive agents but below which their erosive capacity rapidly declines[10].

In this study, we aim to address and reconcile some of these issues by reference to the time-transgressive erosional footprint of the Eurasian ice sheet (EIS) over the last ~100 ka glacial cycle. At its Last Glacial Maximum (LGM) extent, the EIS was the third largest Quaternary ice complex after Antarctica and the Laurentide[11], was in excess of 2500 m thick in places, and had a considerable latitudinal range spanning 49°N to 82°N[12]. The impact of the EIS on long-term landscape development was profound, moulding the North Atlantic passive margin and north-

[1]CAGE − Centre for Arctic Gas Hydrate, Environment, and Climate, Department of Geosciences, UiT The Arctic University of Norway, Tromsø, Norway. [2]Geography Research Unit, University of Oulu, Oulu, Finland. [3]Department of Earth Sciences, University of Gothenburg, Gothenburg, Sweden. [4]ARCEx − Research Centre for Arctic Petroleum Exploration, Department of Geosciences, UiT The Arctic University of Norway, Tromsø, Norway. [5]Department of Earth Sciences, Royal Holloway University of London, Egham, UK. [6]Geomorphology and Glaciology, Department of Physical Geography, Stockholm University, Stockholm, Sweden. [7]Bolin Centre for Climate Research, Stockholm University, Stockholm, Sweden. [8]School of Geosciences, University of Edinburgh, Edinburgh, UK. ✉e-mail: henry.patton@uit.no

western European continental shelf through kilometre-scale denudation, sediment evacuation, and related uplift during repeated Quaternary glacial cycles[13]. Despite this, broad swathes of cold-based inter-fjord uplands and the terrestrial hinterland, including the Baltic Shield, survived multiple glaciations with virtually no modification[14–16]. Recently, these views of a cold-based "relict" landscape have been somewhat challenged by cosmogenic exposure measurements that demonstrate significant glacial modification of high plateaus between the fjords of western Fennoscandia[2,17,18]. Erosion of these high plateau surfaces has been linked to long-term equilibrium line altitudes that yield a distinctive bimodal erosion/elevation distribution due to consequent zonation of cold and warm-based glacier action. This observed bimodal erosive signature is claimed to provide further, nuanced, evidence for a glacial buzzsaw mechanism[19].

We apply a transient 3D thermomechanical ice-sheet model, constrained by terrestrial and offshore empirical data, to determine the spatiotemporal patterns and rates of glacial erosion across Northwest Eurasia. The model is based on a first-order solution of the Stokes equations governing ice flow to compute the free evolution of ice mass balance, thermodynamics, flow, isostatic adjustment, and geometry of the EIS across a 10 km finite-difference domain[12,20]. The model is coupled to a modern-day reference climate via a distributed temperature index scheme based on mean (1950–2000) seasonal temperature and precipitation patterns and is forced by perturbations in these fields scaled to the NGRIP δ18O record and eustatic sea-level change (Fig. 1). The key variables of ice thickness, thermal structure, internal deformation, basal motion and meltwater production are

integrated in discrete time-steps from the last interglacial (Eemian or Marine Isotope Stage - MIS5e) at 123 ka BP through to the present day (Supplementary Fig. 1). This time-transgressive model framework enables us to investigate and contextualize the transient climatic, topographic, geological, and glaciological controls on landscape development throughout the entire last glacial cycle.

In the transient model, subglacial erosion is implemented through an ice flux relation[21,22] that is scaled and calibrated against back-stripped trough mouth fan (TMF) sediments and [10]Be cosmogenic nuclide measurements (see "Methods") across three primary geological provinces that characterize the Eurasian domain: Shield, Younger Basement, and Platform, delimited by rock type and age (Fig. 2a). Shield areas are typically represented by crystalline igneous and high-grade metamorphic basement rocks, and form relatively low relief within tectonically stable areas. The Baltic Shield of Sweden, Finland and northwest Russia is composed mostly of Archean and Proterozoic plutonics (quartz-bearing) >1 Ga. The largest group of younger (<1 Ga) metamorphic rocks that comprise the Younger Basement province are within the mountains of Eurasia (Norway, Scotland, Svalbard) and belong to the Caledonian Orogeny that formed during the collision of Baltica and Laurentia. Subsequent collapse of the Caledonian and Uralian (Russia) orogenic belts has exposed windows of Precambrian-aged rocks.

The Permian-Triassic transition marked the start of progradation of a major sedimentary system into the Barents Sea basin, leading to the infilling of several kilometres of sediment and shallow-marine terrigenous deposits, facilitated by what was the largest delta plain in Earth's history[23]. Early Cenozoic break-up between the Greenland and

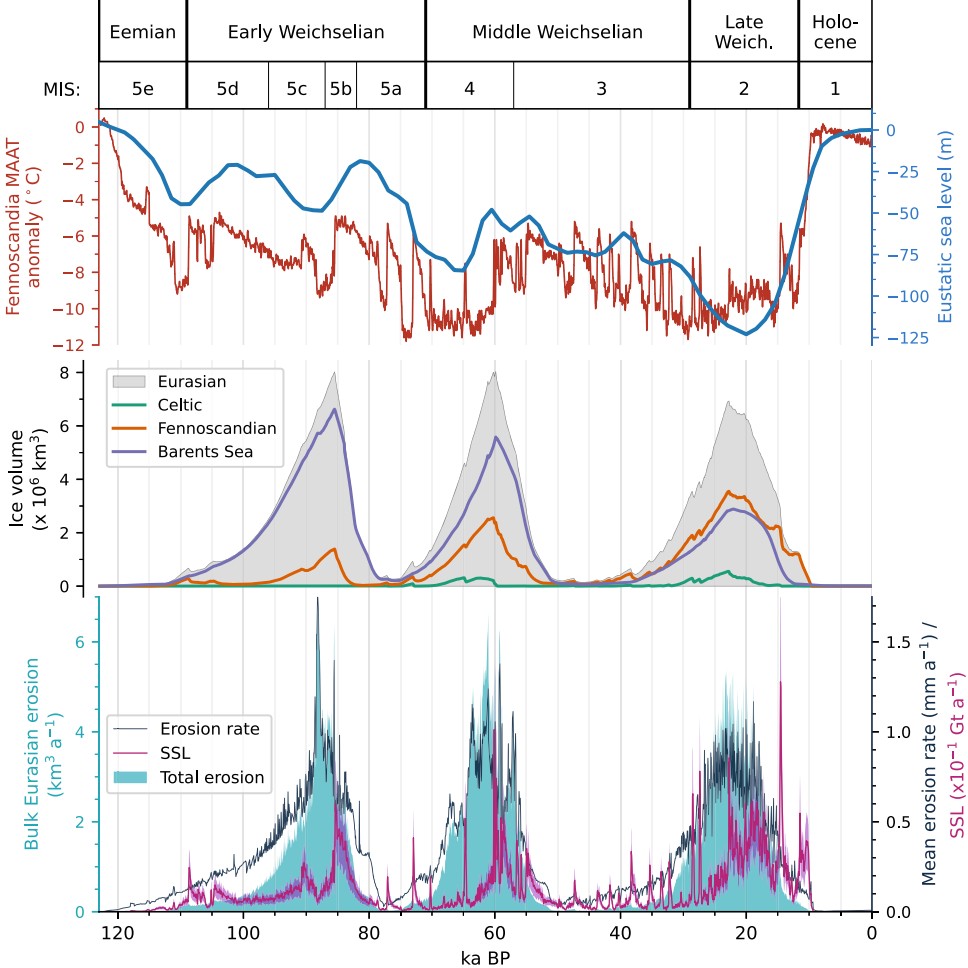

**Fig. 1 | Modelled outputs of the Eurasian ice sheet over the last glacial cycle.** Evolution of the climate (NGRIP[101]) and eustatic sea level[63] forcing curves, ice volume, erosion model outputs, and suspended sediment load (SSL) estimates from 123 ka BP to present day.

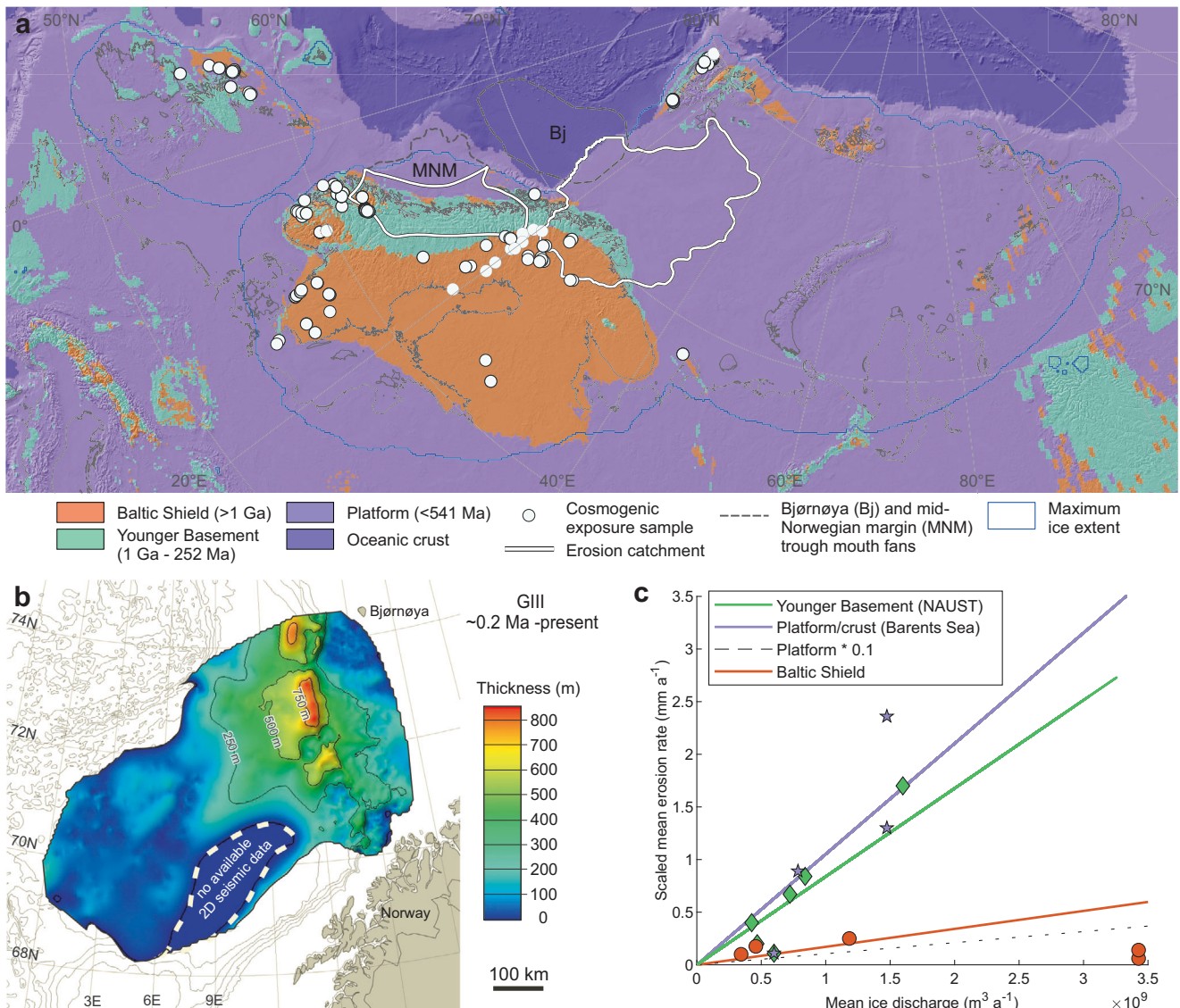

**Fig. 2 | Geological provinces and erosion modelling constraints. a** Geological provinces of the Eurasian domain, delimited by age and rock type according to the IGME 5000[102] and GLiM datasets[103]. Cosmogenic exposure samples taken from bedrock ($n = 249$) were used to constrain glacial erosion rates over the Shield province (Supplementary Data 1). Trough mouth fan catchments, defined from ice model outputs, delimit the areas for backstacking glacial sediments in the Younger Basement (mid-Norwegian Margin) and Platform (Bjørnøyrenna) provinces. **b** Thickness map of the GIII glacial unit in the Bjørnøya trough mouth fan, derived from 2D seismic profiles (Supplementary Fig. 2) and used to constrain erosion rates in the Platform province. For the time-to-depth conversion, we applied an interval velocity of 1.97 km s⁻¹ [28]. **c** Rates and gradients of glacial erosion within distinct geological provinces scaled against mean modelled ice discharge through the last glacial cycle (<123 ka) (see "Methods"). Previously published empirically derived data points (filled markers) are sourced from catchments across the Eurasian domain (Supplementary Fig. 3; Supplementary Table 2).

Eurasian plates formed major rift basins in the present-day Northeast Atlantic and Arctic Ocean, within which further siliclastic sediments, evaporites, and thick carbonate-rich sedimentary sequences accumulated to develop the Platform province. Across the Barents Sea shelf today, an upper regional unconformity separates variously dipping Mesozoic sedimentary rocks from an overlying sequence of glacigenic sediments. In the central part of the Barents Sea the thickness of this Quaternary sediment cover is normally less than a few tens of metres, whereas thicker sequences of 0–300 m occur near the continental slope and in the southeast[24,25].

## Results
### Empirical erosion scaling constraints
We derive empirically constrained erosion scaling laws for Platform and Younger Basement provinces using volumetric calculations of glacigenic sediment deposits in the Bjørnøya and mid-Norwegian

TMFs, respectively (Fig. 2c). Their glacial catchments represent the largest erosional source areas for their respective geological provinces, and as such, best represent the range of glacial dynamics relevant for scaling erosion rates more broadly.

The Bjørnøyrenna catchment dominated ice drainage from the western Barents Sea shelf, extending inland towards ice domes over Fennoscandia, the central Barents Sea and Svalbard. Outcropping strata within this catchment is largely characterized by the thick sedimentary succession (e.g., shales and sandstones) that infilled the Barents Sea basin during the Mesozoic and generally have similar mechanical properties to the Mesozoic sedimentary rocks found farther south in the Eurasian domain. Towards the southern and western Barents Sea shelf, the platform is divided into minor highs and sub-basins mainly influenced by salt tectonics (e.g., Samson Dome)[26]. From a synthesized 2D seismic dataset of 580 profiles covering the Bjørnøya TMF (Supplementary Fig. 2) − the largest glacigenic sediment deposit

within the Arctic − we identify five seismic sequences of sediment deposition within the uppermost glacial (G) seismic unit, GIII, bounded by the R1 regional (R) unconformity at its base (nomenclatures follow the regional framework set out by Faleide et al.[27]). By applying previously defined velocity constraints[28] we estimate a total GIII sediment volume of 62,105 km³ (Fig. 2b). Assuming the age of the R1 unconformity (GIII base) is 0.2 Ma[29,30], we split this sediment volume according to the three major advances of the EIS during the last glacial cycle, to derive a total eroded bedrock volume of 37,263 km³ for the last glacial cycle. The density of the source sedimentary sediment rocks is assumed to be comparable to the erosional products (compaction ratio = 0%)[31].

A comparable large-scale glacigenic sediment depocentre − the Naust Formation − can be found on the mid-Norwegian margin, which built-up over the past 2.7 Ma and allowed the continental shelf to prograde seaward by as much as 150 km. Within the youngest 'T' sequence (0–200 ka), 16,300 km³ of sediment has been deposited, equivalent to 13,000 km³ of bedrock given a compaction ratio of 20%[32] due to the varying lithologies of the source areas (i.e., crystalline rocks). In lieu of more precise chronologies for the glacigenic deposits found offshore, we conservatively estimate 50% of this bedrock volume − 6500 km³ − was eroded since the Eemian, taking into account a reduced ice presence on the mid-Norwegian margin during the early stages of the last glacial cycle (Supplementary Fig. 1). The source catchment for each TMF is drawn using depth-averaged ice velocities through the last glacial cycle (Fig. 2a; Supplementary Fig. 3), yielding areas of erosion of 594,000 km² and 216,800 km², for the Bjørnøya and mid-Norwegian margin TMFs respectively (Supplementary Table 1), and from which scaling factors can then be determined (Fig. 2c).

With limited insight on volumes of glacigenic sediment transported from Shield areas, we use a suite of 249 cosmogenic [10]Be measurements from bedrock samples (Fig. 2a) to calibrate a bulk scaling factor that best fits the duration of ice cover over the last 2.7 Ma according to the LR04 stack[33] (a compilation of benthic $\delta^{18}O$ records from 57 globally distributed sites) and patterns of modelled ice discharge (Supplementary Fig. 4a). For each sample a scaling factor is determined that yields a glacial erosion rate and sample depth history compatible with the measured [10]Be concentration. A Monte Carlo approach with 10,000 iterations is used to estimate positive and negative uncertainties. The median scaling factor value of these data points is applied in the erosion scaling law, yielding a fit approximately one order-of-magnitude less than that for the Younger Basement and Platform sectors (Fig. 2c).

For all provinces we acknowledge there is no explicit accounting for the initial evacuation of pre-Weichselian sediments, which would lead to a variable overestimation of the long-term erosion rates. This is partially compensated through the implementation of a lower threshold for erosion at depth-averaged velocities <10 m a⁻¹, which is of particular relevance for areas near the former ice-divide where pre-Weichselian sediments are widely reported[34]. Furthermore, the TMF sediment volumes of Weichselian age used for back-stripping in Younger Basement and Platform provinces also implicitly contain a proportion of sediments that were eroded during older glaciations but were later deposited off-shelf during the Weichselian. However, the time-scale for the initial evacuation of these shelf sediments is ambiguous, which today on the Barents Shelf range in thickness from 0 to 300 m[25], and ~0 to 20 m across Shield areas[35]. In highly lubricated zones of fast-streaming ice in West Antarctica, contemporary geophysical observations of soft sediment erosion rates of up to 1 m a⁻¹ [36] demonstrate that the active deposition, reworking and removal of subglacial sediments overlying bedrock can be extraordinarily rapid. Contrarily, pockets of pre-Weichselian sediments preserved through the last glacial cycle in some ice-marginal environments in Svalbard[37]

and Denmark[38] suggest that erosion was not always pervasive beyond the core areas of the ice sheet.

## The glacial erosional footprint

The empirically constrained model analysis confirms that the EIS had a colossal erosional footprint that was directly responsible for the net removal of ~126,500 km³ of bedrock during the last glacial cycle. Over 82% of this total eroded volume is sourced from sedimentary strata associated with the Platform geological province, which in total accounts for 71% of the formerly glaciated domain (Table 1), including most of the offshore sectors (Fig. 2a). Shield areas, despite composing 21% of the domain, source only 8% of erosional products (Table 1). Across all sectors, the bulk pattern of erosion is primarily concentrated along the main arterial discharge pathways of the EIS, with largest removal depths of 80–190 m within the troughs of the Barents Sea (Supplementary Fig. 4b), corresponding to rates of 1.4–5.2 mm a⁻¹ (Fig. 3). Zones of suppressed erosion, characterized by rates less than 0.1 mm a⁻¹, are sustained across upland and divergent flow zones of Fennoscandia, the Barents/Kara seas, Great Britain, and Ireland. Over 19% of the entire domain, and 55% of the land area above 1000 m a.s.l., experienced less than 2 m net erosion, coincident with relict zones of landscape preservation in upland areas and those of extreme cold in the Kara Sea and Taimyr Peninsula (Supplementary Fig. 4b).

When aggregated over the glacial cycle, long-term patterns of erosion correspond well with independent (bulk) empirical estimates, particularly within marine-terminating source areas (Supplementary Table 1). Comparison with a recent source-to-sink sediment budget for the Baltic Sea Basin[39] reveals potential overestimation of the predicted long-term glacial erosion rate in this sector of the EIS. This could reflect model limitations in reproducing ice streaming through this basin, thereby leading to excessive ice fluxes and inferred catchment size, or possibly equally due to methodological issues in the estimation and reconstruction of the erosion rates reported. Further, the discrepancy can also be indicative of the varied protective role of previously worked sediments deposited in the source area during earlier glaciations, which our model does not account for. Lastly, the Neoproterozoic and early Palaeozoic sedimentary rocks found in the Bothnian and Baltic basins are fundamentally different in terms of their mechanical properties (hardness and fracture density) when compared to the softer Mesozoic sedimentary rocks found elsewhere across other platform areas in the domain[39], highlighting the local complexities associated with geological controls when modelling potential erosion at the continental/ice-sheet scale.

Over the short-term, results derived using the model will be susceptible to some variation due to choices in parameter values that impact ice flow and discharge, such as the basal sliding scheme utilized or in modifications to the ice flow law that enhance internal deformation (see "Methods"). A comprehensive suite of sensitivity experiments has previously been conducted to investigate the impact of incremental changes in internal and external parameters on our model results[12]. For this study, to assess the sensitivity and impact of modelled ice-sheet dynamics on the resulting erosion patterns, we systematically perturb the primary determinant of ice flow − the rate-factor multiplier ($A_0$) − in a series of experiments conducted during peak glaciation at the LGM (Fig. 4). During this 2000-year period, glacial erosion rates across the mid-Norwegian margin broadly follow similar trends, but with significant excursions during short-lived (sub-millennial) episodes of high ice flux. Subsequent rescaling of the Younger Basement erosion law to account for these varying ice fluxes, yields a sensitivity difference amounting to ~1 mm of net erosion during such high ice-discharge episodes, lending confidence both to the approach and the reconstructed patterns of erosion. Ultimately though, the bulk erosional products are finitely constrained over the glacial cycle, thereby effectively limiting the impact of uncertainties in

**Table 1 | Breakdown of predicted glacial erosion**

| | Weichselian (<123 ka BP) | | | | | Post Last Glacial Maximum (<22.7 ka BP) | | | |
|---|---|---|---|---|---|---|---|---|---|
| | Total erosion (km³) | Mean erosion (m) | Mean erosion rate (mm a⁻¹) (<123 ka) | Mean glacial erosion rate (mm a⁻¹) (±1σ) | Erosion <2 m (% total area) | Total erosion (km³) | Mean erosion (m) | Mean glacial erosion rate (mm a⁻¹) (±1σ) | Erosion <2 m (% area glaciated since LGM) |
| Platform (71.6% area) | 104,127 (82.3%) | 20.24 | 0.16 | 0.79 (0.76) | 19.29 | 17,510 (72.3%) | 4.90 | 0.84 (0.91) | 41.83 |
| Younger Basement (7.6%) | 11,975 (9.5%) | 22.03 | 0.18 | 0.46 (0.48) | 10.96 | 2550 (10.5%) | 4.90 | 0.52 (0.55) | 36.44 |
| Shield (20.9%) | 10,414 (8.2%) | 6.95 | 0.06 | 0.24 (0.18) | 21.99 | 4,141 (17.1%) | 2.78 | 0.28 (0.21) | 46.06 |
| Total Eurasian IS | 126,518 | 17.60 | 0.14 | 0.65 (0.70) | 19.23 | 24,203 | 4.34 | 0.66 (0.79) | 42.45 |
| Younger Basement low elevation (<550 m a.s.l.) | 9699 (81.0%) | 25.41 | 0.21 | 0.59 (0.51) | 9.61 | 2050 (80.4%) | 5.67 | 0.64 (0.61) | 32.80 |
| Younger Basement high elevation (≥550 m a.s.l.) | 2276 (19.0%) | 14.05 | 0.11 | 0.16 (0.16) | 14.14 | 500 (19.6%) | 3.14 | 0.25 (0.25) | 44.70 |

Volumes of eroded bedrock are calculated by geological province, as well as from high- and low-elevation sites within the Younger Basement regions (Norwegian and Scottish Caledonides).

model degrees of freedom and specific parameter choices on our results beyond sub-millennial timescales.

## Discussion

### Spatiotemporal controls on glacial erosion

Geology and climate are conventionally regarded as the key controls on glacial erosion. Attempts to link glacial erosion rates with precipitation or latitudinal proxies have yielded statistically robust results, though consistent comparison of glacier types is a key consideration[5,7,40]. Analysis of our modelled subglacial erosion patterns across the Northwest Eurasian domain fails to yield statistically significant support for a synoptic-scale link between glacial erosion and latitude (Fig. 5b). However, this negative result does not entirely preclude a latitudinal link, and may reflect the all-encompassing nature of an ice sheet's subglacial footprint compared to that of individual glaciers, where it is possible for the most dominant glacial catchments to span multiple climatic zones or even geological provinces. Instead, our results consistently support a nuanced hierarchy of interdependent environmental controls based on lithology, topographic relief and climate, all of which are critically modulated by the continuously evolving thermomechanical configuration of the ice sheet.

On the inter-regional scale, we find that geology can provide a key control on erosion. For example, erosion rates across the Baltic Shield are an order-of-magnitude lower than adjacent provinces (Table 1), ostensibly due to rock hardness and fracture density that limit abrasion and plucking[41,42]. Persistent cold-based conditions coupled with high overburden loads across northern Fennoscandia also combine to protect the substrate, limiting the impact of meltwater and groundwater over-pressurization[43]. Similarly, restricted erosion across low-lying platforms of northern Russia and the Kara Sea reflect cold-based subglacial conditions that correspond with limited entrainment of perma-frozen clayey sediments by basal ice[44].

Contrasting sedimentary and crystalline bedrock lithologies have also been implicated as a distinguishing control on bulk glacial erosion, with rates doubling between the Bjørnøyrenna catchment and the mid-Norwegian shelf, respectively (Supplementary Table 1)[31]. While bedrock properties associated with different lithologies can exert a primary control on the dominant erosional mechanism, such as abrasion (soft, thick-bedded, wide joint spacing) versus plucking (hard, thin-bedded, narrow joint spacing)[42], generally we find no significant difference in erosional efficacy between these two provinces once our model results are normalized for the duration of ice-sheet occupancy (Fig. 2c).

Sustained ice streaming routed via the main arterial troughs across the Barents Sea reflects both the broad ice-sheet configuration and warm basal conditions, with the distinct absence of lateral barriers promoting efficient mass discharge from the ice-sheet interior.

Observations of multi-phase flow trajectories of the Bjørnøyrenna ice stream since the LGM attest to the spatiotemporal variability and occurrence of significant flow switching[45] that would act to limit the development of glacial overdeepenings[46]. In contrast, the deep, narrow fjords of western Norway invoke transverse shear zones that act to stabilize and direct ice flow, suppressing lateral development of outlet discharge from the hinterland to the shelf break. Whilst this deep linear erosion of the Norwegian coast and inner shelf has yielded significant sediment volumes, these topographic steering feedbacks can ultimately act to limit erosion as the long profiles of beds beneath highly erosive glaciers tend towards steady-state angles opposed to and steeper than the overlying ice surface[47]. Across sufficiently steep retrograde slopes, negative feedbacks related to channel closure through glaciohydraulic supercooling will reduce the bedload transport capacity and ultimately favour sediment deposition over bedrock erosion[46,48].

On $10^{4-5}$ year (ice age) timescales, climate a priori controls glacial erosion since ice-sheet mass balance, volume and dynamic activity ultimately scale to snow accumulation. We find that peak erosion of 5–7 km³ a⁻¹ occurs during three phases: 88 ka, 61 ka and 23 ka BP, coeval with maximum EIS volume and positive mass balance (Fig. 1). Erosion patterns also deviate by ±1 mm a⁻¹ over centennial timescales in response to abrupt warming. Such transient behaviour - when the cryospheric system is perturbed from its equilibrium state - arises from abrupt external forcing, though can also result from internal ice-dynamical related switches due to thermomechanical reconfiguration.

Following such arguments, the erosion patterns described above could, reductively, be ascribed exclusively to climatic amelioration and its impact on ice dynamics. For example, the mean EIS erosion rate by province was 6–17% higher during the last 8 ka deglaciation compared to the bulk mean over the glacial cycle (Table 1). Yet, further analysis of the spatiotemporal erosion patterns reveals that this difference of up to ±1.6 mm a⁻¹ results from dynamical reorganization during the asynchronous build-up of the EIS to its LGM extent (Fig. 5c). The shifting eastward extent from 23-19 ka BP in response to ice-divide migration[12,49] intensified ice flow and erosion within the central Barents Sea and Russian sectors later than elsewhere. Subsequent warming did promote meltwater runoff and subglacial discharge during deglaciation, which further enhanced erosion rates of these late-stage ice streams[50,51], but climate itself was not the underlying control.

The uplands of Fennoscandia have traditionally been considered relict landscapes preserved by cold-based ice. However, the extent to which glacial action has been limited to superficial scouring[3,52] or pervasive down-wasting[2,18,19] across these upland zones is debated. Our subglacial erosion results demonstrate that high-elevation sites (≥550 m a.s.l.) within the Younger Basement province account for ~19%

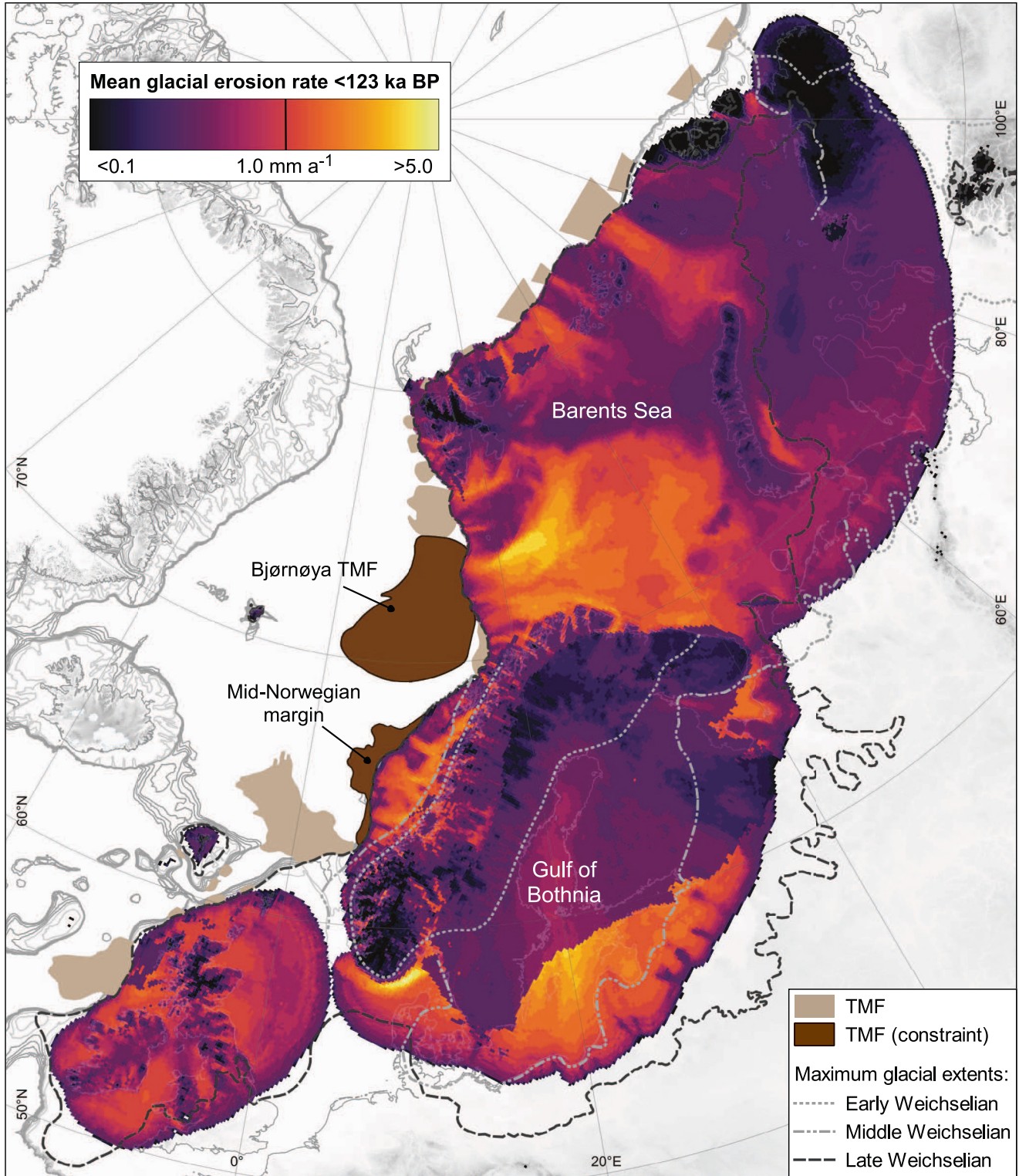

**Fig. 3 | The erosional footprint of the EIS.** Subglacial erosion over the last 123 ka, normalized for ice occupancy. Empirical-based limits of glaciation and the trough mouth fans (TMF)[104, 105] used to constrain glacial erosion rates in Platform (Bjørnøyrenna) and Younger Basement (mid-Norwegian margin) provinces are highlighted.

of the total sediment flux to offshore depocenters (Table 1). Previous partitioning of glacigenic sediments transported off the mid-Norwegian shelf broadly correspond to between 10 and 45% derived from high-elevation plateaus[2,19]. Our analysis indicates limited support for a bimodal signature of erosion across western Fennoscandia over the entire glacial cycle. However, we do find an internal thermomechanical control on erosion across high-elevation plateaus during phases of deglaciation. For example, mean erosion rates increase from 0.16 to 0.25 mm a$^{-1}$ after the LGM at elevations above 550 m a.s.l. (Table 1; Fig. 5c). Such time-transgressive impacts, masked by spatially complex patterns of erosion and preservation, illustrate how the substrate and evolving thermodynamic regime interact to

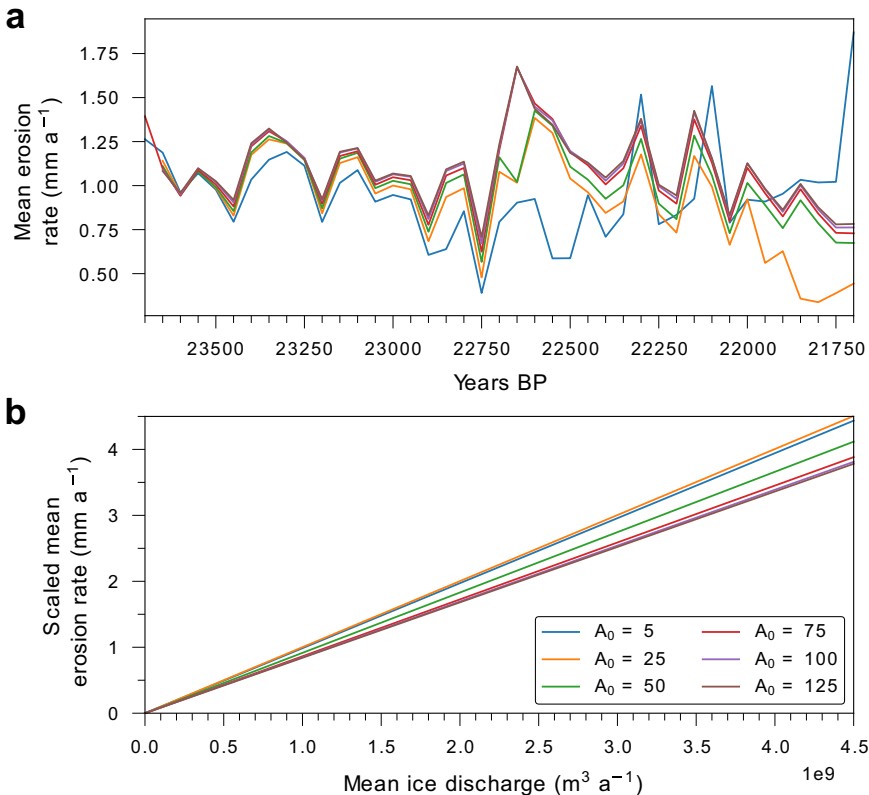

**Fig. 4 | Potential variation in modelled erosion rates. a** Mean erosion rates across the mid-Norwegian margin catchment (Fig. 2a) during a 2000-year window around the Last Glacial Maximum, through perturbations of the $A_0$ deformation enhancement factor in the ice model. The value used in our optimum experiment is 75, and erosion laws are kept consistent across each experiment. **b** Erosion scaling laws for the Younger Basement province derived from each sensitivity experiment assuming a constant volume of 472.26 km³ (derived from the optimum experiment) was eroded during the 2000-year timeframe in (**a**).

drive episodes of enhanced upland erosion. Our results also reveal a relatively modest increase in bulk erosion at ~800 m a.s.l. which might be interpreted as evidence of a bi-modal signature and support for the buzzsaw hypothesis, but rather, we find these patterns relate to the hypsometric distribution of relief, and not to any intensification of glacial erosion at this elevation (Supplementary Fig. 5).

Our analysis demonstrates that bulk patterns are principally dominated by focused and selective erosion below 550 m a.s.l. (Fig. 5a). While Baltic Shield erosion is generally modest over the last glacial cycle, deep erosion of Younger Basement and Platform provinces at rates of up to 5.2 mm a⁻¹ during transient phases account for the bulk sediment volume and stratigraphic architecture found within TMFs bounding the shelf edge. This vigorous, deep regional incision from Shield margins, along with episodes of enhanced erosion driven by abrupt disequilibrium in the climate-cryosphere-landscape system, help contextualize contemporary process studies that support extreme rates of glacial erosion[7,53] when these cryospheric systems are rapidly transitioning to new equilibrium states.

**Suspended-sediment fluxes**

Glacial erosion and transport of sediments, including those suspended in proglacial runoff, loaded numerous TMFs distributed along the marine shelf edge offshore Scotland, Norway, and the Barents Sea (Fig. 3). The quantification of suspended-sediment concentrations adjacent to modern-day glaciers and ice sheets is also used as an indirect proxy to derive short-term rates and trends of subglacial erosion[53,54]. To examine these relatively ephemeral studies within a broader geological context we utilize our own time-transgressive framework to derive and contrast potential suspended sediment (i.e., ignoring bedload) fluxes and trends throughout the last glacial cycle.

Bulk fluxes are derived by scaling contemporary process measurements from the Greenland ice sheet (see "Methods") to meltwater runoff discharge calculated from the mass balance algorithm of the ice-flow model. Assuming a continual source of subglacial sediments, fluxes from catchments along the North Atlantic and Arctic Ocean margins peaked during phases of rapid climate warming and deglaciation, attaining a maximum of 12.78 ± 5.1 Gt a⁻¹ (Fig. 1). This resulted in a total delivery of ~50,000 km³ of sediment into the ocean system during the last glacial cycle, yielding 125 m of net aggradation along the continental shelf-break. The maximum suspended sediment flux, associated with the collapse of the EIS during the Bølling-Allerød warming ~15 ka BP when surface melting of the ice-sheet surface was most intense and had extended northwards to affect the Eurasian Arctic too[20], is an order of magnitude greater than contemporary rates from the Greenland ice sheet, and equivalent to the present contribution from all rivers globally[54]. The impact on downstream marine ecosystems during this episode would have been substantial, limiting primary productivity in the pelagic zone by reducing light availability in the water column and increasing the far-field supply of bioavailable micronutrients to promote phytoplankton blooms, yielding a major carbon sink[55].

While subglacial erosive efficacy broadly scales to EIS flow dynamics, linked to the progressive influence of ice-stream activity through ice-sheet evolution[56], subglacial mobilization of those sediments as suspended loads is phase-lagged by millennia and driven by episodes of abrupt warming and the associated rapid increases in meltwater and runoff delivered to the bed (Fig. 1). This provides a direct analogue to the contemporary response and deglaciation of the Greenland ice sheet under abrupt 21ˢᵗ century warming[57]. Record meltwater fluxes accessing the ice-bed interface due to anomalously warm and wet summers[58], coupled with enhanced dynamics as interior

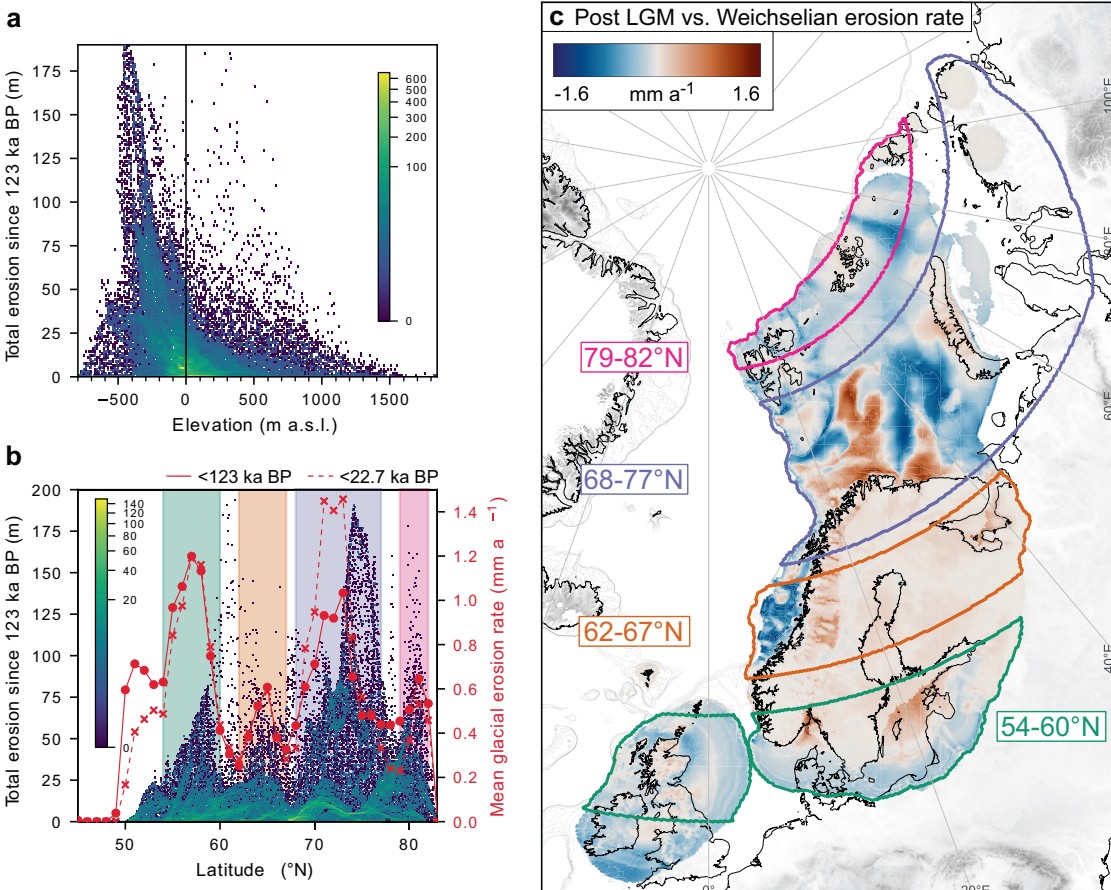

**Fig. 5 | Spatial patterns of modelled erosion. a** Point density cloud of total glacial erosion by elevation during the last glacial cycle. **b** Total and mean glacial erosion for the last glacial cycle by latitude. Coloured banding are areas of relatively intense erosion (see **c**) over the entire cycle and the last deglaciation (<22.7 ka BP).

**c** Changing intensity of glacial erosion during deglaciation (<22.7 ka BP) compared to the long-term mean. Positive values indicate increasing rates of erosion during deglaciation, and negative values indicate decreasing erosion rates.

regions of the Greenland ice sheet transition to wet-bedded regimes, promote the mobilization of new, subglacial sequences of sediment[59,60] by more extensive and intense basal drainage, with resulting suspended loads suggesting erosion rates orders-of-magnitude above the considered long-term norm[53,61]. Such insights into the transient nature of these processes − where increased melt runoff under abrupt warming accesses new zones of the subglacial environment with increased vigour to mobilize previously reworked and deposited glacial sediments − highlights a cautionary tale when upscaling field-measurements from specific point sources on short (contemporary/process) timespans to inform long-term landscape evolution models[5,7].

In summary, our analyses demonstrate that the EIS, which at its maximum spanned over 5500 km of temperate through extreme polar conditions from 49 to 82°N, was an extremely effective agent of erosion. Despite this, its spatiotemporal signature was also highly heterogenous and transient, with its erosion potential determined by an interacting hierarchy of environmental and internal controls: climate, lithology, topographic relief, and the dynamically evolving, thermomechanical ice-sheet configuration. The time-transgressive perspective presented here reveals the complex and highly nuanced impact of glacial erosion on long-term landscape evolution, which, at a continental ice-sheet scale, rudimentary environmental proxies such as latitude or climatic regime fail to capture.

Though our analysis discounts the case for any deep, uniform glacial erosion of the Fennoscandian shield, the internal dynamic modulation demonstrated by our transient analysis provides support

for the view that ice sheets can be able or feckless erosive agents in the same place at different times[62]. Moreover, the highly selective subglacial erosion across shield margins, along with episodes of extreme rates driven by abrupt disequilibrium within the climate-cryosphere-landscape system, provide essential context to contemporary process studies that imply exceptionally high rates of erosion − or rather the mobilization of reworked subglacial sediments − as ice sheets respond to increased meltwater and runoff fluxes to the basal environment and deglaciate under abrupt climate warming.

## Methods
### Ice-flow model description
The three-dimensional thermomechanical model applied and its associated boundary conditions are of the same derivation as those used by Patton et al.[12,20] to model the Late Weichselian Eurasian ice sheet (EIS) (Supplementary Table 3), with several modifications made to make it suitable for running over full glacial timescales. In brief, the ice-flow model is a first-order approximation of the Stokes equations and includes longitudinal (membrane) stresses that become increasingly important across steep gradients in topography and motion. The model is integrated forward through time on a finite-difference grid through perturbations in climate (NGRIP $\delta^{18}O$) and eustatic sea level[63] (Fig. 1).

Ice velocity, composed of internal deformation and basal motion, is determined through flow and sliding relations. Internal deformation uses an adaption of Glen's flow law[64] that includes an empirical flow enhancement (aka softening) coefficient, used to encompass the

effects of crystal anisotropy and impurities on bulk ice deformation[65]. Modifying the strain rate towards softer ice tends to increase ice-flow speeds. Basal sliding is related to the basal shear stress through a Weertman-style sliding law, that is introduced at sub pressure-melting temperatures using an exponential decay function[66,67]: at 0.75 K below the pressure-melting point, sliding is 0.47 of its value at the pressure-melting point. While this classical sliding law is based on the assumptions of a glacier flowing over hard bedrock, its applicability can break down in subglacial environments characterised by widespread cavitation or 'soft'/till-laden beds that induce a Coulomb-type regime where basal shear stress is proportional to the effective pressure. Coulomb friction is thus more representative of sliding near the grounding line. Real-world sensitivity analyses regarding the choice of friction law on grounding line dynamics reveals a varied transient response, with Weertman laws systematically predicting the smallest changes in ice volume above flotation[68], though the exact nature of the effect on ice dynamics due to parameter choices in sliding laws can be highly region-dependent[69].

Surface mass balance is determined by a positive degree-day (PDD) scheme[70], and derives total melt from integrated monthly positive temperatures. Both temperature and precipitation adjust to the evolving ice-sheet surface through applied lapse rates derived from multiple-regression analyses of meteorological observations at a resolution of 1 km from the WorldClim database (Version 1.4)[71]. To account for the large variations in climate regime across the Eurasian domain, regional reference climates and associated forcing are tuned independently for each of the three major ice-sheet accumulation centres (from north to south: the Barents Sea, Fennoscandia, and Great Britain and Ireland). An additional mass-balance term incorporated is the net water–vapour flux to and from the ice-sheet surface − a predominant component of ablation in cold continental settings where humidity can be very low[72].

Calving losses at marine-terminating margins are coupled to relative sea level using a standard empirical function relating the calving flux to ice thickness and water depth[73]. The sensitivity of calving to, for example, variations in ocean temperature and sea-ice buttressing has been controlled spatially and temporally through a depth-scaled calving parameterization[74]. In the absence of explicit calculations of such external feedbacks, this depth-related calving coefficient provides a pragmatic and computationally efficient parameterization for determining mass loss at marine terminating margins of the EIS. The model is applied to a 10 km finite-difference mesh with the inclusion of grounding-line dynamics based on the analytical boundary treatment of Schoof[75] and adapted in 2D by Pollard and DeConto[76], which defines the ice flux at the grounding line as a function of ice thickness linearly interpolated between the adjacent node that brackets floating and grounded ice.

Several modifications were made to the ice-flow model to make it suitable for running over full glacial timescales. The first accounts for insolation variations and their impact on the surface mass balance. This parameter can account for 20–50% of the surface melt anomaly, although its effects are not homogenous over the ice sheet since the contribution of insolation to melt is modulated by the local surface albedo[77]. We apply a simple and tested correction factor to the PDD melt scheme ($M_{corr}$), making it possible to tune the insolation-independent parameters from present-day data, while relying on the insolation term to handle changes in palaeo conditions:

$$M_{corr} = M(S_0) + M_{insol} \qquad (1)$$

$$M_{insol} = a(T,d)(S - S_0) \qquad (2)$$

where $M_{insol}$ is the insolation-related melt, $S$ is the incoming solar radiation at the top of the atmosphere, and $S_0$ is the spatially and

seasonally explicit insolation fixed to present-day levels. The spatially and seasonally varying coefficient, $a$, can be considered to be representative of average local conditions of surface albedo and atmospheric transmissivity for the ice sheet, in effect acting as a substitute for a full surface-albedo model. The dependence on temperature means that $a$ cannot be taken as a constant, but must contain some seasonal variation. To better capture this local variability, $a(T,d)$ is parameterized as a piecewise linear function of the local mean daily temperature ($T$) and the day of the year (d):

$$a = \begin{cases} 0, & \text{if } T \leq T_{min} \\ a_{max}\frac{(T-T_{min})}{(T_{max}-T_{min})}, & \text{if } T_{min} < T < T_{max} \\ a_{max}, & \text{if } T \geq T_{max}. \end{cases} \qquad (3)$$

For simplicity, the values of $a_{max}$ and $T_{max}$ were chosen to be constant[77]. $T_{min}$ − the minimum temperature threshold for melt corrections − was parameterized as a nonlinear sinusoidal function of the time of year between $T_{max}$ in winter and $T_{min,sum}$ in summer:

$$T_{min} = T_{max} - (T_{max} - T_{min,sum})\left(\frac{1 - \cos(2\pi\frac{d}{365})}{2}\right)^p. \qquad (4)$$

The values of $a_{max}$, $T_{max}$, $T_{min,sum}$, and the exponent $p$ used are based on tuning to best match the diagnosed values of $a$ across the Greenland ice sheet[77].

## Ice-flow model experiment design

Ice-modelling experiments were initiated for the Eemian climatic optimum at 123 ka BP to allow small ice caps to equilibrate with Eemian climate. Two important assumptions for these experiments are that the Eemian topography was similar to that of the present-day in terms of isostatic relaxation and relief, and that Eemian interglacial glaciers had reduced to a negligible size.

Manually applied perturbations of climatic patterns, such as major rain shadow effects, were kept consistent with those applied during the optimum Late Weichselian experiments of Patton et al.[12], though the magnitude of these climate gradients were tuned separately for the previous two ice-sheet advances during the last glacial cycle. One major divergence from the original setup was to add a dynamic adjustment to mimic a Kara Sea precipitation shadow by scaling the precipitation gradient with ice-surface elevation over central Novaya Zemlya. The resulting effect freely encourages ice growth over the Kara Sea during phases of ice build-up but prevents runaway ice expansion once the central ice dome thickens.

The aim of these modelling experiments is to produce a glaciologically feasible EIS simulation that fits within the broad geological framework related to ice extension during the last glacial cycle (Fig. 3; Supplementary Fig. 1). The uncertainty associated with the few geophysical constraints for glaciation over this time period means that the envelope within which a potential reconstruction could fit remains open-ended. For example, what was the initial ice topography during the Eemian, and did ice centres always fully retreat during Weichselian interstadials? Although specific questions relating to the evolution of the ice complex over multiple glacial advances cannot currently be answered, the experiment used in this study provides a framework from which insights into longer-term processes can be examined.

Sensitivity experiments of the LGM reconstruction with an essentially identical setup were carried out by Patton et al.[12], highlighting the variation of the maximum ice-sheet configuration to a range of key climate and internal parameter choices. Furthermore, the pattern and rate of retreat across the Eurasian Arctic after the LGM was independently validated and compared with other model reconstructions using a suite of relative-sea level curves from around the Barents Sea[20,78].

## Subglacial erosion scaling laws

The long-term mean erosion rate, $\bar{f}$, is derived by linearly scaling eroded bedrock volume ($B$) with mean ice discharge ($\bar{q}$) (Fig. 2c) and the duration of ice coverage ($t_i$):

$$\bar{f} = \frac{(B | \sum \bar{q} \cdot t_i) \cdot (\bar{q} \cdot t_i)}{t_i}. \tag{5}$$

Eroded bedrock volumes for the Platform and Younger Basement provinces are derived from volumetric calculations of the Bjørnøya and mid-Norwegian margin TMFs, respectively. To tune spatial patterns of long-term erosion to observations of pre-glacial landscape preservation, erosion was limited to areas where depth-averaged velocities are at least 10 m a$^{-1}$. The resulting linear fit between the rate of erosion (mm a$^{-1}$) and local ice discharge within both erosion catchments is thus described (Fig. 2c):

$$f_{YB} = 8.3832 \times 10^{-10} \cdot q - 8.8373 \times 10^{-20}, \tag{6}$$

$$f_{\text{Platform}} = 1.0512 \times 10^{-9} \cdot q + 1.6817 \times 10^{-19}. \tag{7}$$

The widespread preservation of relict glacial-[79–81] and non-glacial landscapes[82–84], and of preglacial weathering remnants on the Fennoscandian Shield, highlights the minimal impact of Pleistocene glacial erosion within central areas of the former EIS[15,85–89]. Semi-quantitative estimates of total glacial erosion on valley floors within central shield areas range from ~2.5 to 50 m[16,85,90,91]. With limited further insight on volumes of glacigenic sediment transported from Shield areas, we use a suite of 249 cosmogenic [10]Be measurements from bedrock samples to calibrate a bulk scaling factor that best fits the duration of ice cover over the last 2.6 Ma according to the LR04 stack[33] and patterns of modelled ice discharge (Supplementary Data 1).

For each sample a scaling factor, $F$, is calculated that yields a glacial erosion rate and sample-depth history compatible with the measured [10]Be concentration. The calculations are based on the glacial erosion calculator glacialE.m version 201912 from the expage cosmogenic nuclide calculators (http://expage.github.io/calculator). Cosmogenic [10]Be production rates are calculated based on the nuclide-specific LSD method[92] with production from muons calibrated against the Beacon Heights depth-profile data[93] and using a global average reference spallation production rate of 3.98 ± 0.22 atoms g$^{-1}$ a$^{-1}$. The time-dependent [10]Be production rate is calculated based on sample location (latitude, longitude, and elevation), topographic shielding, sample depth, bedrock density, and history of ice cover. We assume full shielding from cosmic rays and a zero cosmogenic nuclide production rate through all periods of ice cover. The local ice discharge and ice cover periods from the ice-sheet model are used for the last glacial cycle. Similar to the ice-sheet model, glacial erosion occurs only when depth-averaged velocities are at least 10 m a$^{-1}$. For the time 2.6 Ma to 123 ka, we use the LR04 stack[33] as a proxy for ice cover, with an ice cover cut-off value determined by the modelled total duration of ice cover in the last glacial cycle with linear interpolation of the LR04 cut-off value from 3.2‰ (123 ka ice cover) to 5.0‰ (0 ka ice cover). For a particular cosmogenic nuclide location, we use the average ice discharge from the last glacial cycle for all ice cover periods prior to 123 ka. Because the last duration of exposure to cosmic rays is commonly critical for the [10]Be concentration, we adjust the timing of the last deglaciation to match the reconstructed deglaciation from DATED-1[94]. The deglaciation adjustment is only applied to the ice cover history (shielding from cosmic rays), and it does not affect the total glacial erosion derived from the ice-sheet model. For the ice-free periods we assume no interglacial erosion. For twelve samples collected from sites below the highest shoreline in Sweden, the production rate is adjusted for the water depth calculated from the uplift model by Påsse and

Daniels[95], with uplift following pre-LGM ice cover periods based on the duration of ice cover and the uplift history following the last deglaciation. We then calculate the central value of F and use a Monte Carlo approach with 10,000 iterations to estimate positive and negative uncertainties, applying the reported [10]Be concentration uncertainty, the reference production-rate uncertainty, [10]Be decay constant uncertainty[96,97], a 5% uncertainty for bedrock density, a 1 ka uncertainty for the DATED-1[94] deglaciation age, a 10% uncertainty for the vertical uplift for samples located under the highest shoreline, a 0.1‰ uncertainty for the LR04 ice cover break value for the glaciations prior to the last glacial cycle, and a one-sided positive uncertainty of 0.005 mm a$^{-1}$ for the interglacial erosion rate. For full details of the calibration, we refer to the provided supplementary code (Supplementary Data 2). Using the median scaling factor value of these data points we produce a linear glacial-erosion rule with a fit almost one order-of-magnitude less than that for the Younger Basement and Platform sectors (Fig. 2c):

$$f_{\text{shield}} = 1.7062 \times 10^{-10} \cdot q. \tag{8}$$

## Uncertainties in estimating glacial erosion

The reworking of older glacial and interglacial sediments during glacial advances, in addition to the volume of sediments not transported to the TMF depocenters, are potential sources of bias in our back-stacking calculations. For example, in the present interglacial, glacigenic sediments across the Barents continental shelf today range in thickness from 0 to 300 m[98], and the pattern of sediment thicknesses across the Baltic Shield exhibits a pronounced spatial pattern consistent with a range of 0–20 m[35]. Similarly, the presumption for totally efficient removal of sediments from the subglacial system will likely compound this uncertainty[52], although their evacuation can proceed rapidly once active[51], at rates of up to 1 m a$^{-1}$ beneath streaming ice[36].

The use of cosmogenic nuclide data as a proxy for constraining erosion rates, while arguably more precise, is limited by the shallow depth of cosmogenic-nuclide production in rock and, consequently, a high sensitivity to glacial erosion depths during the last ice cover period (erosion deeper than the depth of production cannot be constrained) and durations of exposure/shielding during the post-glacial period. Modest, local variation in glacial erosion depths can also yield nearby surfaces with highly varying cosmogenic-nuclide concentrations[41,99]. Samples for cosmogenic nuclide analysis are often sampled away from valley/basin floors (mostly because of sediment fills), which could bias estimates of erosion. For example, the sampling of plucked bedrock surfaces to yield reliable deglaciation ages by avoiding cosmogenic-nuclide inheritance[99], leads to unconstrained erosion estimates in a landscape where glacial erosion otherwise was minimal, creating a complex and inconsistent patchwork of constraining point data. Similarly, samples can also be taken to establish non-glacial erosion rates, such as on blockfields or tors where geomorphic evidence indicates negligible glacial erosion.

Modelled patterns and rates of glacial erosion correspond well with available empirical records of long-term denudation along the western seaboard of the Eurasian continent (Supplementary Table 1), and are qualitatively in-line with the inferred zonation of erosion based on the spatial pattern of Fennoscandian Quaternary deposits[52]. Volume-based sedimentation rate estimates derived from shield sources, which fed sink areas south and east of the EIS, are, however, poorly constrained, and generally limited by uncertainty in the initial terrestrial sediment volumes. One estimate for Quaternary sediments deposited by the EIS yields a mean thickness of ~55 m across the Eastern European Plain, increasing to 84 m within the LGM domain[100]. With a source area representing erosion of the Shield in Finland and the Platform in the eastern Baltic, Gulf of Finland and East European Plain, these estimated sediment volumes provide an order-of-magnitude indication of glacial erosion on the scale of up to tens of metres over the last 2.7 Ma,

aligning with mean modelled estimates of glacial denudation of 6.95 m (0.06 mm a$^{-1}$) across Shield areas for the last glacial cycle (Table 1).

## Suspended sediment load estimates

Suspended sediment fluxes were scaled across the Eurasian domain using observations from the modern-day Greenland ice sheet. We incorporate the analyses of Overeem et al.[54], who measured suspended-sediment concentrations from 160 rivers (17% of the total meltwater flux) using Landsat7 Enhanced Thematic Mapper and Earth Observation-1 Advanced Land Imager (ALI) imagery and in-situ sampling. Sediment concentrations were analyzed over the active river-discharge summer season for 14 years, yielding a dataset honouring the large variation in discharge events and highly nonlinear nature of sediment transport. The median suspended-sediment concentration examined was found to be 992 mg l$^{-1}$. Direct upscaling of these observations yielded a total ice-sheet contribution of $1.28 \pm 0.51$ Gt a$^{-1}$, associated with a mean yearly runoff of 418 km$^3$ (RACMO2.3 1999–2013)[54]. An uncertainty of 40% in the total estimate stems from the errors in the reconstructed suspended-sediment concentration from all 160 river outlets, and the estimated error in the modelled runoff (20%). By determining riverine suspended-sediment concentrations through a first-order erosion model linked to glacial dynamics instead, an alternative estimate for the total sediment load from Greenland amounts to $0.892 \pm 0.374$ Gt a$^{-1}$ (with a relative error of 42%)[54].

We directly scale the former estimated suspended-sediment flux to meltwater fluxes calculated within the mass balance budget of the ice-sheet model through the last glacial cycle, assuming a consistent sediment concentration through time. Sediment load fluxes were further partitioned according to broad ice-sheet scale catchments defined in Patton et al.[20].

## Data availability

The bedrock $^{10}$Be cosmogenic data analyzed and used for calibration of the ice sheet / glacial erosion model is provided in Supplementary Data 1. Raw data are derived from the expage compilation of glacial $^{10}$Be and $^{26}$Al data (http://expage.github.io/).

## Code availability

Matlab code used for analyses of the cosmogenic exposure datasets are provided in Supplementary Data 2.

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

## Acknowledgements

This research is a part of the Centre for Arctic Gas Hydrate, Environment, and Climate and was supported by the Research Council of Norway through its Centre of Excellence funding scheme (grant 223259) and the *Akademia* Programme at Equinor. Ice modelling experiments were performed using High-Performance Computing services through Sigma2 (project #NN9465K). AH gratefully acknowledges an Arctic Interactions fellowship at the University of Oulu funded by the Finnish Academy of Sciences and an Arctic Five Professorship.

## Author contributions

H.P. contributed ice-sheet modelling outputs and analyses, wrote the original manuscript, oversaw the project, and with A.H. developed the study, the ice model code, and further edited and developed the text. J.H. analyzed and simulated the cosmogenic exposure histories. N.A. and M.W. analyzed seismic data to contribute trough-mouth fan sediment volumes. A.P.E.L., A.P.S., A.M.H., M.W., D.E.S., J.K., and K.A. contributed with the synthesis and integration of observational and geological datasets from marine and terrestrial sectors with ice sheet modelling results, and advised on the scope of the study. H.P. and N.A. prepared the figures. All authors openly discussed ideas, extensively commented on, and edited the manuscript.

## Funding

## Competing interests

The authors declare no competing interests.
