## [Peer Review File · Nature Communications]

The extreme yet transient nature of subglacial erosionREVIEWER COMMENTS

Reviewer #1 (Remarks to the Author):

The paper combines numerical, geochronological, geophysical and sedimentological data to estimate subglacial erosion under the Eurasian Ice Sheet (EIS) during the last glacial cycle. The results suggest that ice sheets are highly efficient agents of erosion and sediment re-distribution while the erosion intensity varies strongly in time and space. In general, the derived erosion rates are in the acceptable range and the conclusions are intuitively plausible, although one could expect a higher role of local lithology in modulating the erosion and denudation. The paper is clearly presented, properly illustrated, and the conclusions (although possibly not particularly spectacular) are mostly sound (but see below).

The conclusions of the paper including the numerical reconstruction of erosion refer to the whole EIS but the presented data come entirely from its northern and—to a lesser extent—central parts. Critically, the volumes of bedrock eroded from what is referred to as Platform province are based on the sediment volumes in the Bjørnøyrenna and Mid Norwegian trough mouth fans (TMF). However, these fans were not capturing the sediment mobilized from the southern part of EIS, which also belong to the Platform province. This potentially generates a large error in the empirical erosion scaling constraints. Furthermore, the sediment volumes in the TMFs only represent a fraction of the eroded bedrock because the finer-grained sediment has been carried away in suspension. The issue of the suspended sediment flux is touched upon in the Methods, but the problem remains.

The ice-flow model itself is technically sound. The authors refrain from using the simpler, but unrealistic "shallow ice approximation", but actually use a better description of ice physics (the "first-order approximation" of ice stress). However, they use a "Weertman tomb-stone" style parameterization of basal sliding, and this is somewhat old-fashioned. The sliding physics control the initiation and magnitude of ice streaming, so this may be an important point with regards to erosion rates in fast-moving areas. Schoof, Zoet and Iverson all published more realistic sliding models in the period 2010-2020 (see the references at the end of these comments), which are based on principles such as Coulomb friction, cavitation, and hard/soft beds. The authors should at least acknowledge the limits of their simpler sliding model, and reflect on what the choice may implicate. Furthermore, they should do some form of uncertainty analysis on the ice-flow model, in order to assess how important the parameters in Table S1 are to the outcomes of the study.

The authors relate total ice flux at a given point with erosion-rate data sourced from cosmogenic nuclides measurements in order to compute the erosion rate. This is different from a process-based model, which considers the physics involved in the problem. Therefore, the erosion model applied here is strictly empirical, obscuring insight to transitions in erosional processes across the ice sheet (e.g. abrasion vs. plucking, bed lithology, fractures). Secondly, the tuning parameter in the empirical erosion model shows that erosion efficacy is drastically reduced in the Baltic Shield outside of the mountainous areas (Fig. 3A, C). As their model scales erosion with ice flux, it may simply be a consequence of ice-fluxes being too large in the flat parts of the Baltic Shield, or ice may be too slow in the other areas.

In sum, the major critique points are:

1. Lack of consideration of the southern parts of the EIS in estimating volumes of eroded sediment in the Platform province based on North-Sea TMFs.
2. The choice of sliding model is not consistent with recent findings.
3. The empirical fit between ice flow and erosion masks insight into processes at play.
4. The lack of uncertainty quantification of the ice-flow model itself makes it impossible to assess the criticality of parameter choices.

Minor points:

1. Put more "beef" in the Abstract; it reads too much like a "story telling" now.
2. Insert references in figure captions (e.g., Fig. 1) and text (e.g., lines 37-39) wherever data from other sources are used.
3. Line 110: "Fig. S1".

4. Be consistent in abbreviating "year" as "a" or "yr", not both.

5. Fig. 1: "Middle Weichselian" not "Mid Weichselian".

References:

Schoof, C. 2005. The effect of cavitation on glacier sliding. *Proceedings of the Royal Society A* 461, 609–627.

Zoet, L.K. & Iverson, N.R. 2015. Experimental determination of a double-valued drag relationship for glacier sliding. *Journal of Glaciology* 61(225), doi: 10.3189/2015JoG14J174.

Zoet, L.K. & Iverson, N.R. 2020. A slip law for glaciers on deformable beds. *Science* 368, 76–78.

Reviewer #2 (Remarks to the Author):

The ms by Patton et al. focuses on the styles of subglacial erosion underneath ice sheets, using the Eurasian Ice Sheet (EIS) as a case study. Given the nature of the topic (understudied, yet important for many different fields within earth sciences and beyond) and the fact that the EIS was the third-largest ice sheet throughout much of the Pleistocene, the work itself is of global significance and suitable for the broad readership of *Nature Communications*.

I should preface this review with a caveat: I am not a modeler and could thus not assess/scrutinise the code or approach. I am also not an expert in CRN dating. However, as a terrestrial sedimentologist I am a bit more familiar with the subjects of erosion and sedimentation, and I will focus my review on points relating to this area in the hope this is useful to the authors.

The ms is well written and accessible, and the figures are very well drafted. The findings are certainly presented in such a way that the main arguments of the authors can be followed with ease. I list a few cases where I have had questions and/or problems identifying whether the authors have thought about some of the caveats (beyond those stated in the methods section). I list my observations under the following headings:

1. Representativeness of the Bjørnøyrenna TMF

The authors use seismic evidence from one TMF off the W coast of Norway to bracket sedimentation rates throughout the Pleistocene and to back-calculate erosion rates in the fan's catchment that are then normalised across the ice sheet. This approach is a good one, and I find it very plausible. It is also a very nicely-done integration of marine and terrestrial datasets and approaches, which is highly commendable.

One question that I find is not at present addressed explicitly in the text is that of how representative this catchment (and thus fan) may be of the entire EIS? Is it a small catchment? Is the bedrock in it more erosive (weaker) than that found in other catchments? How variable is the bedrock in this catchment? And, related to point 2 below, how much sediment cover could have been expected covering the bedrock trough prior to the underlying bedrock even being 'attacked' by each ice sheet throughout the Pleistocene?

2. Role of sediment overlying bedrock

In various parts of the ms (e.g. lines 161ff.) you discuss the role of bedrock type (including joint spacing etc.) on erodibility and the efficacy of subglacial erosion. While I fully follow these arguments (and those in the methods section as well), I do have a bit of a problem with the way the sediments are treated, or rather the way they seem to be ignored in the calculations. I don't think this has been done deliberately, but maybe more information on this would satisfy the curious and critical reader.

Sediments would have most likely covered the bedrock (the erosion of which you aim to quantify) at the beginning of each glacial cycle (as a result of the preceding glaciation, deglaciation as well as paraglacial processes operating on those sediments in the preceding interglacial). While a few words on this are lost in the methods (lines 426ff.), these focus on the methodological uncertainties of extracting quantified values from the impressive CRN dataset from bedrock surfaces. I am by no means suggesting you redo any of the analyses (or, probably worse, include depth-profile CRN data), but what I would suggest is this:

Any sedimentary cover overlying the bedrock needs to be removed by the ice (especially when

considering greater erosion underneath thicker, temperate ice in glacial troughs or on trough flanks and potentially shoulders; cf. line 426f.). In some cases, we know that sediment thickness can exceed several deka-metres (e.g. tunnel valleys), in some overdeepened valleys almost as much as on fjord floors, i.e. a kilometre and more (cf. Preusser, F., Reitner, J.M. & Schlüchter, C. Distribution, geometry, age and origin of overdeepened valleys and basins in the Alps and their foreland. *Swiss J Geosci* 103, 407–426 (2010). <https://doi.org/10.1007/s00015-010-0044-y>). While some of this sediment will be preserved throughout multiple glacial cycles and added to under favourable conditions, you currently only deal with this by stating areas near EIS ice divides where we know erosion was less efficient. However, I struggle to see how thick areas of sediment in the lowland areas (and those between ice divides and those lowland areas near the margins) would NOT affect an averaged erosion rate calculated from bedrock alone, i.e. with those sediments that make up the vast majority of the ice sheet bed in ~30-~50% of the area covered by it. Certainly, if those sediments would have been stripped away at least partially, it would make the erosion rates you calculate more conservative, i.e. they could potentially underestimate the true erosive capability of the entire EIS. I have no idea how much of an underestimate this would give, but I would suggest putting this caveat into the main discussion rather than hide it in the methods – it is a central problem! One way to get a quick estimate, if the authors wanted to include a number/bracketing values, might be to use a sediment thickness map (base of the Quaternary) that is available for several countries. I also note that such eroded sediment would be included in all of the TMFs, including the one you use to derive catchment erosion rates. This means that data from this catchment could also fairly easily be included in any calculations for the entire EIS and perhaps give a more reliable estimate of net erosion at ice-sheet scale?

At any rate, I would strongly recommend you include this caveat explicitly rather than leaving it in its implicit form when talking about limited erosion at ice divides. Perhaps you could include a few references to work from the EIS that has shown partial preservation of such sediments in areas further away from the ice divides such as southern and central Sweden, and outside the mountains?

3. Suspended sediment load

I am not quite sure what these data add to the overall argument. They do seem to be an add-on, but one that can be done without. My main argument here is that abrasion clearly produces rock flour that would be transported as suspended load, but this would most likely not make it from the source (point of abrasion) to the margin in most cases, but be deposited from suspension along the flowline of the ice sheet (e.g. in subglacial lakes or mixed in with other sediments such as subglacial traction till). By only including suspended load, you also ignore (at least explicitly as this is not stated in the text) the contribution of abrasion versus plucking in bedrock erosion. In any case, I would recommend adding a sentence or two to the methods and discussion. At present, the rationale for including suspended sediment load escapes me, unfortunately.

Figures

These are of a very high quality throughout, and my comments are fairly minor, but hopefully they help clarify the visual message.

Fig. 2: "Celtic" (ice sheet): Sorry to say, but this is a historically-nonsensical misnomer: The Celts actually inhabited much of continental Europe all the way to S Spain and into Eastern Europe way beyond any pre-Weichselian ice limits! The accepted term for this sector of the EIS is the British-Irish Ice Sheet (BIIS). You also mention "British" somewhere in the text.

The figure needs a legend that decodes the TMFs.

Why are the TMFs in the NW sector of the BS sector delineated by unnatural-looking straight lines? They have presumably been mapped (i.e. their exact planforms could be represented) like the others?

Could you please include a stippled line to show the maximum extent of the ice sheet during the last glaciation, so that the areas you present erosion envelopes for can be put into its full geographical context, in particular to the ice margins?

Fig. 3: A: Include the abbreviations from the caption in a legend. Also decode the white dots in the same legend. The location of panel B is missing from A (include a rectangle or dot) - it took me a

while to locate B in A.

B: I like this figure! One plea: wouldn't it be better and more intuitive to convert the thickness measurement from metres per second into true thickness in metres? I am suggesting this, because that way the figure could also be more easily included for both research and teaching purposes.

C: The scales to me seem odd: I would expect "(...)" for the units rather than a "/"

Fig. 4: "Point density cloud" rather than "map" would be more intuitive to me.

Table 1: Is there a reason "Y.B." has dots as the only abbreviation in the ms?

"Y.B. plateau (≥ 550 m a.s.l.)": Not all sites above 550 m qualify as plateaux, neither in Scandinavia, Scotland nor elsewhere. This to me is a gross oversimplification that should either be clarified or removed altogether. The most un-ambivalent and uncomplicated way of doing this would be to replace this with "Y.B. high elevation (≥ 550 m a.s.l.)".

Fig. S3: A: the unit in the top right-hand corner is unclear to me (typo?). Do you mean volume or area here?

General observations

- There are a few minor, but still irritating problems with the correct hyphenation of compound adjective-noun adjectives versus normal adjective and noun constructions (for example, lines 33-34: it should be "low-relief continental shields", but never "high-latitudes").
- 35: "the drivers of and controls on"
- 98 (and elsewhere): "GIII" – this is not specified anywhere else in either text or figures. Please decode and be more specific.
- 108: "T formation" – this is not specified anywhere else in either text or figures. Please decode and be more specific.
- 112: "areas" – should be "volumes"?
- 147: "we do find evidence". This is very interesting! What evidence is this specifically? And what does it indicate? Please provide more detail here on the hierarchy of controls and type of evidence you have (is it 'hidden' in one of the figures and could be highlighted, for example? Does it come out of particular model runs?). Otherwise, if you're unable to, I suggest you drop this general statement, because it is a bit of a mean teaser.
- 157: "perma-frozen clays": Is it really just clays or better to state "sediments"?
- 161: should be "primary controls"
- 251: decode GrIS in previous line
- 254: would "insights into the transient nature of these processes" be clearer? I struggle with the idea of an insight being transient (i.e. fleeting)
- References: There are a few incomplete bibliographic entries: numbers 39, 57, 75 and 79 need attention.

I hope these comments are helpful in rounding the message of this highly interesting and topical ms even further.

Please do not hesitate to get in touch directly with me as well if you want to discuss any of my comments.

Sven Lukas, LU

Reviewer #3 (Remarks to the Author):

Review of " The extreme yet transient nature of subglacial erosion" (NCOMMS-22-06881)

Reviewer: Simon Cook, University of Dundee, UK

Summary

I was grateful for the opportunity to review this manuscript by Patton et al. Determining the controls on glacial erosion rates is an important and interesting subject, with many fundamental issues still unresolved despite many decades of effort. The authors present a wide-ranging and hard-won set of results, including marine geophysical, geological and geochronological data, and outputs from the modelling of ice sheet dynamics over the last glacial cycle. The sheer scale of the Eurasian Ice Sheet, spanning a huge latitudinal range, allows the authors to test some key hypotheses about what controls glacial erosion beneath ice sheets. They arrive at the conclusion that ice sheet thermodynamics is the dominant factor that determines glacial erosion rates, and that other factors that have previously been held up as the main controls (climate/latitude, lithology, etc.) are less significant. I'm broadly supportive of this effort and its conclusions, and I think the sheer spatial and temporal scale under consideration here, as well as the depth of analysis make it very impactful, and hence a good fit with Nature Communications. I have some broader comments followed by minor issues. I hope the authors find these helpful in refining their work, though it is already very well written and seems to have been put together with great care.

- Article title. One to consider, but not a deal-breaker by any means... I wonder if the title of the article is truly reflective of the key message of the paper. I like the title, and I can see the appeal of using words like 'extreme', but it strikes me that this title is more reflective of the observations of the paper rather than its actual conclusions (i.e. that ice dynamics regulates erosion rates). I think people are already aware that erosion can be variable; they may not be so aware of the importance of ice thermodynamics, which is perhaps where the real novelty is here. Finally, the maximum erosion rate found in this study (5.2 mm/year) is not the extreme limit of glacial erosion – Alaskan tidewater glaciers have been found to erode at 10s of mm per year.
- The role of latitude, climate and "latitude/climate" in glacial erosion. Getting straight to the point, I don't think what I discuss below will need much to fix, but I think it is important that the authors are clear about what they are testing when it comes to the latitudinal/climatic control on glacial erosion because it is one of the central hypotheses that they test in this study. The authors tend to equate latitude with climate ("latitude/climate"), which is a very crude equivalence. Ultimately, I think the authors have tested the relationship between latitude and glacial erosion rate, and not necessarily climate (by which I mean temperature and precipitation) and erosion rate.

As we demonstrated in Cook et al. (2020), latitude is probably only a very crude proxy for climate, and even then only for temperature, and not precipitation. In our dataset, we lumped all sorts of glaciers together and found only a very weak relationship between latitude and climate. Koppes et al. (2015) did find a relationship, but only because they isolated latitude as a controlling variable by sampling only tidewater glaciers across a wide latitudinal range. So I think the results of Cook et al. (2020) support your results of only weak correlation between latitude and erosion rate.

Around L144-149, a direct comparison is made between the results of this study and the results of previous studies. That's important to do of course, but I think there is a bit of missing nuance here. Previous work (Koppes et al., 2015, Anderson et al., 2016, and Cook et al., 2020 are cited in the preceding sentence of the manuscript) has examined the relationship between latitude, and to a lesser extent, climate, for valley glaciers. Actually, to be fair, Cook et al. mixed in a few ice sheet outlet glaciers, which on reflection is perhaps a bit unhelpful in this context. Nonetheless, should we really expect ice sheet erosion to be limited by the same climatic/latitudinal constraints as valley glaciers? It's hard to tell exactly from the images here (nice as they are), but it looks like some of the glaciers draining the EIS crossed very large latitudinal ranges themselves – possibly crossing geological boundaries, possibly passing through different climates (temperature and precipitation regimes), perhaps snow falling in one place and feeding an ice stream that does most of its erosion hundreds of km away, and so on. My point being that this is on a totally different scale and context to the sorts of glaciers that have been used in the past to determine sliding/climate/latitude-erosion relationships. Maybe latitude works better for carefully selected

valley glaciers (Koppes et al., 2015; Anderson et al., 2016) than it does for ice sheets (this study) or less-choosy selections of glaciers (Cook et al., 2020). I just wonder whether the authors need to be a little less forthright in their conclusions and acknowledge these nuances (see for example L266).

Even if the authors consider that they are examining the relationship between climate and erosion rate, I still wonder if what they really mean is temperature, or latitudinally controlled temperature. And as we showed in Cook et al. (2020) for (mostly) valley glaciers, precipitation is also very important. But I take the authors' overall point that ice sheets are far more able to set their own rhythm of glacial erosion.

Minor points, typos, etc.

L36-39. Since you're giving examples, I think you need some references here.

L49. What exactly is meant by "kilometre-scale denudation"?

L100. Maybe I've missed this somewhere, but how do you split the total volume of the GIII unit (62,105km³) into three glacial advances and end up with a value of 37,263km³ for the most recent glacial cycle? Glacial cycle duration? Some geophysical and geochronological evidence?

L108 – what is the "T formation"? I know there are citations, but I think it needs to be explained in this paper.

L112 – areas of erosion with units of km³. Should these be km²?

L112 – "respectively" is used, but it's not clear in what order the TMF/catchments are being considered from the text here.

L117 – what is the "LR04 stack"? First time this has been mentioned and it's not clear what we're dealing with. I'm guessing some geophysically-derived sediment facies from one of these TMFs?

L126, though a general point. Here, it is stated that the total eroded volume has been 126,518 km³. That seems super-precise! Elsewhere, you've quoted areas to the nearest thousand, and then other volumes (L99 and L102) to the nearest km³. Given the uncertainties in geophysical measurements, dating, ice dynamics and so on, I'm wondering whether these very precise numbers are really warranted.

L135. I'm not sure you should use the term 'British Isles' given that the Celtic Ice Sheet covered the island of Ireland too.

L143. "correlate relationships" seems an odd phrase.

L143. Not sure the tone is quite right here. "...climate or latitudinal proxies alone have met with limited success". I'd argue there has been progress rather than limited success.

L165. This diagram (Fig 3C) doesn't show the result alluded to in the text here.

L176-177. This point about negative feedback loops could be better explained. Rather vague. I guess what you mean is that retrograde slopes induce glaciohydraulic supercooling, which leads to low sediment transport capacity in subglacial streams, hence deposition of a protective till layer that limits erosion. If that's the case, then I guess you don't want to use so many words to say all that, but I really think you should at least cite Alley et al. (2003) Nature (Stabilizing feedbacks in glacier bed erosion, I think). But what I'm not clear on here is that this passage of text is referring to linear erosion of fjords, whereas what you are discussing in this final sentence is about overdeepenings. I know fjords commonly have overdeepenings, but you don't actually say that, so it seems like a logic step is missing here.

L214. Shouldn't this refer to Fig S5?

Fig 1. Does the y-axis label for SSL need to be coloured purple to match the line colour? I realise this might not be straightforward to do in the code, and possibly more hassle than it's worth.

L225 and whole section on suspended sediments. Can we be clear here: what do you mean by suspended sediment loads (SSL)? Is this a catch-all term for all sediment exported in subglacial and proglacial streams, including bedload? Or is it only suspended sediment, in which case these numbers may only represent minimum sediment export estimates because they ignore bedload? As you're probably aware, bedload transfer is rarely measured in sub-/pro-glacial systems; most glacial erosion estimates are based on suspended sediment loads measured at a meltwater portal. But this ignores bedload, or only considers it using some fudge factor. I'm curious to know what is considered in this study, or what the assumptions are.

L254-6. I think this is an important point about the extrapolation of modern glacial erosion rates and sediment fluxes to longer-term studies. Others have made this same point, including Koppes et al. (2015), Cook et al. (2020), Jaeger and Koppes (2016), and so on. Perhaps a citation or two could be added here.

L290. "Version 1.4" currently in superscript, but probably should be placed in brackets in regular font, before the citation number.

L318 and L321. Inconsistent font style for term, "a".

L325. Suggest reword to, "For simplicity, the values of...to be constant⁵⁸"

L333. "Ice experiments" seems an odd/vague term.

L699. data are

Fig 3. Do you need r^2 and p -values for your lines of best fit?

We would like to thank the three reviewers for their time and effort in providing constructive and detailed appraisals of the manuscript. We have incorporated and responded to each of their suggestions, and have below indicated our specific point-by-point response to each comment and query.

Reviewer #1:

The paper combines numerical, geochronological, geophysical and sedimentological data to estimate subglacial erosion under the Eurasian Ice Sheet (EIS) during the last glacial cycle. The results suggest that ice sheets are highly efficient agents of erosion and sediment re-distribution while the erosion intensity varies strongly in time and space. In general, the derived erosion rates are in the acceptable range and the conclusions are intuitively plausible, although one could expect a higher role of local lithology in modulating the erosion and denudation. The paper is clearly presented, properly illustrated, and the conclusions (although possibly not particularly spectacular) are mostly sound (but see below).

The conclusions of the paper including the numerical reconstruction of erosion refer to the whole EIS but the presented data come entirely from its northern and—to a lesser extent—central parts. Critically, the volumes of bedrock eroded from what is referred to as Platform province are based on the sediment volumes in the Bjørnøyrenna and Mid Norwegian trough mouth fans (TMF). However, these fans were not capturing the sediment mobilized from the southern part of EIS, which also belong to the Platform province. (Andersen et al., 2004). Furthermore, the sediment volumes in the TMFs only represent a fraction of the eroded bedrock because the finer-grained sediment has been carried away in suspension. The issue of the suspended sediment flux is touched upon in the Methods, but the problem remains.

The ice-flow model itself is technically sound. The authors refrain from using the simpler, but unrealistic "shallow ice approximation", but actually use a better description of ice physics (the "first-order approximation" of ice stress). However, they use a "Weertman tomb-stone" style parameterization of basal sliding, and this is somewhat old-fashioned. The sliding physics control the initiation and magnitude of ice streaming, so this may be an important point with regards to erosion rates in fast-moving areas. Schoof, Zoet and Iverson all published more realistic sliding models in the period 2010-2020 (see the references at the end of these comments), which are based on principles such as Coulomb friction, cavitation, and hard/soft beds. The authors should at least acknowledge the limits of their simpler sliding model, and reflect on what the choice may implicate. Furthermore, they should do some form of uncertainty analysis on the ice-flow model, in order to assess how important the parameters in Table S1 are to the outcomes of the study.

The authors relate total ice flux at a given point with erosion-rate data sourced from cosmogenic nuclides measurements in order to compute the erosion rate. This is different from a process-based model, which considers the physics involved in the problem. Therefore, the erosion model applied here is strictly empirical, obscuring insight to transitions in erosional processes across the ice sheet (e.g. abrasion vs. plucking, bed lithology, fractures). Secondly, the tuning parameter in the empirical erosion model shows that erosion efficacy is drastically reduced in the Baltic Shield outside of the mountainous areas (Fig. 3A, C). As their model scales erosion with ice flux, it may simply be a consequence of ice-fluxes being too large in the flat parts of the Baltic Shield, or ice may be too slow in the other areas.

In sum, the major critique points are:

1. Lack of consideration of the southern parts of the EIS in estimating volumes of eroded sediment in the Platform province based on North-Sea TMFs.

While this would of course be useful exercise to consider these North Sea fans, there is an issue in that the ice flow model does not reproduce well the complex interaction of the Celtic and Fennoscandian ice sheets in the North Sea domain. It is a well-known (paleo) glaciological problem and has not yet been solved satisfactorily by transient ice-sheet models. As such, the dominant eroding agent in this sector – the Norwegian Channel ice stream – is not fully represented and use of these TMFs in our approach here would not be effective.

We take this comment on-board about the consideration of the southern parts of the EIS though, and address this by extending comparison of the modelled vs empirical erosion rates in Table S3 with the Baltic Sea Basin (Hall and van Boeckel, 2020). These catchment comparisons now account for 29% of the modelled domain. Some further sentences have also been added in the section ‘The glacial erosional footprint’, mentioning the differences in this sector and the potential role of sediments in protecting the substrate.

2. The choice of sliding model is not consistent with recent findings.

We agree that implementing a different sliding will yield a degree of variation in the predicted ice flow, and thus the erosion rates, though we would expect the patterns of erosion to remain broadly similar within source catchments. We have added a paragraph to the model description with more comprehensive detail regarding ice flow, and reflection on the sliding law used.

3. The empirical fit between ice flow and erosion masks insight into processes at play.

The tuning parameter for the erosion law in Shield areas is derived using data from 249 cosmogenic nuclide measurements widely spread around the province (Fig 3A), and captures various slow to fast ice-flow regimes. The scaling relationship is compared to other sites in the Shield province where long-term erosion estimates have been estimated (Fig 3C), which shows that across contrasting areas of ice-streaming onset and beneath the central ice divide (Figure S4), this order-of-magnitude difference between the various provinces appears quantitatively valid.

Our new comparison with another source-to-sink budget for the Baltic Sea basin (for point 1) however appears to show our modelling results overestimate erosion in this catchment. Your suggestion that the ice model is overestimating the ice flux around the Baltic may be correct, related to fundamental limitations of the type of ice model that can be applied over such palaeo timescales, but it could also reflect the protective role of old glacially eroded sediments. Lithological variation could also be a further factor, with the local Neoproterozoic and Palaeozoic sedimentary rocks fundamentally different in terms of their mechanical properties when compared to the softer Mesozoic sedimentary rocks that characterise the Barents and UK shelves. We have now brought these discussion points into the text under the section ‘The glacial erosional footprint’.

4. The lack of uncertainty quantification of the ice-flow model itself makes it impossible to assess the criticality of parameter choices.

The ice model used to create the EIS reconstruction through the Weichselian glacial cycle is an extension of the LGM experiments modelled by Patton et al. (2017, 2016), wherein sensitivity analyses of some of the model parameters in Table S1 were carried out. For example, reducing A_0 (deformation enhancement) can lead to a generally thicker ice sheet (by up to 6%) due to 'stiffer'/slower-moving ice. For the LGM deglaciation scenario (Patton et al., 2017) the time-transgressive thickness predictions were then validated using a GIA model to compare the resulting relative sea-level curves with observations from sites around the modelling domain, with the 'best-fit' experiment thus chosen accordingly. We now make mention of these details of this independent validation in the ice-model description.

We have also included in this study now some further sensitivity assessment of the ice model and its impact on erosion rates within a catchment during a 2000 yr period surrounding the LGM. The Naust Formation catchment of the Younger Basement is chosen, representative of both 'shielded/protected' and deeply eroded landscapes. Perturbations of the deformation enhancement factor show some variations around a broadly consistent temporal trend, with mean differences of up to 1 mm a⁻¹ during phases of highest discharge during the LGM. Details on this and a new Figure 5 have been incorporated at the end of the results section.

Minor points:

1. Put more "beef" in the Abstract; it reads too much like a "story telling" now. Some phrasing has been removed/reworded.
2. Insert references in figure captions (e.g., Fig. 1) and text (e.g., lines 37-39) wherever data from other sources are used. Additional citations have been added
3. Line 110: "Fig. S1". Added '.'
4. Be consistent in abbreviating "year" as "a" or "yr", not both. Changed 'yr's to a.
5. Fig. 1: "Middle Weichselian" not "Mid Weichselian". Changed

References:

- Schoof, C. 2005. The effect of cavitation on glacier sliding. *Proceedings of the Royal Society A* 461, 609–627.
- Zoet, L.K. & Iverson, N.R. 2015. Experimental determination of a double-valued drag relationship for glacier sliding. *Journal of Glaciology* 61(225), doi: 10.3189/2015JoG14J174.
- Zoet, L.K. & Iverson, N.R. 2020. A slip law for glaciers on deformable beds. *Science* 368, 76–78.

Reviewer #2:

The ms by Patton et al. focuses on the styles of subglacial erosion underneath ice sheets, using the Eurasian Ice Sheet (EIS) as a case study. Given the nature of the topic (understudied, yet important for many different fields within earth sciences and beyond) and the fact that the EIS was the third-largest ice sheet throughout much of the Pleistocene, the work itself is of global significance and suitable for the broad readership of Nature Communications.

I should preface this review with a caveat: I am not a modeler and could thus not assess/scrutinise the code or approach. I am also not an expert in CRN dating. However, as a terrestrial sedimentologist I am a bit more familiar with the subjects of erosion and sedimentation, and I will focus my review on points relating to this area in the hope this is useful to the authors.

The ms is well written and accessible, and the figures are very well drafted. The findings are certainly presented in such a way that the main arguments of the authors can be followed with ease. I list a few cases where I have had questions and/or problems identifying whether the authors have thought about some of the caveats (beyond those stated in the methods section). I list my observations under the following headings:

1. Representativeness of the Bjørnøyrenna TMF

The authors use seismic evidence from one TMF off the W coast of Norway to bracket sedimentation rates throughout the Pleistocene and to back-calculate erosion rates in the fan's catchment that are then normalised across the ice sheet. This approach is a good one, and I find it very plausible. It is also a very nicely-done integration of marine and terrestrial datasets and approaches, which is highly commendable.

One question that I find is not at present addressed explicitly in the text is that of how representative this catchment (and thus fan) may be of the entire EIS? Is it a small catchment? Is the bedrock in it more erosive (weaker) than that found in other catchments? How variable is the bedrock in this catchment? And, related to point 2 below, how much sediment cover could have been expected covering the bedrock trough prior to the underlying bedrock even being 'attacked' by each ice sheet throughout the Pleistocene?

The Bjørnøyrenna catchment represents one of the largest to have covered a predominantly Platform geological province, and represents well the broader Barents Sea basin geology. In the Barents Sea, platform areas are extensive and relatively consistent in their 'erodibility' – outcropping strata largely comprise Mesozoic shales and sandstones, and which are largely similar to the platform province on the UK shelf. Further information on this is now provided on this platform geology in the geological setting and initial discussion of the Bjørnøyrenna catchment.

We also provide new information on the depth to URU – the interface between Quaternary sediments and bedrock – which typically varies from 0 to 10s of metres in central areas, and up to 300 m nearer the shelf edge. We suggest in the methods that the erosion/evacuation of these soft sediments could have proceeded rapidly, based on observed measurements of rapid (1 m a^{-1}) soft-sediment erosion in Antarctica today. There is some discrepancy in the mechanical properties with platform areas in the Baltic Sea basin compared to the Bjørnøyrenna catchment, which we make reference to with new data in Table S3 and in the section 'The glacial erosional footprint'.

2. Role of sediment overlying bedrock

In various parts of the ms (e.g. lines 161ff.) you discuss the role of bedrock type (including joint spacing etc.) on erodibility and the efficacy of subglacial erosion. While I fully follow these arguments (and those in the methods section as well), I do have a bit of a problem with the way the sediments are treated, or rather the way they seem to be ignored in the calculations. I don't think this has been done deliberately, but maybe more information on this would satisfy the curious and critical reader.

Sediments would have most likely covered the bedrock (the erosion of which you aim to quantify) at the beginning of each glacial cycle (as a result of the preceding glaciation, deglaciation as well as paraglacial processes operating on those sediments in the preceding interglacial). While a few words on this are lost in the methods (lines 426ff.), these focus on the methodological uncertainties of extracting quantified values from the impressive CRN dataset from bedrock surfaces. I am by no means suggesting you redo any of the analyses (or, probably worse, include depth-profile CRN data), but what I would suggest is this:

Any sedimentary cover overlying the bedrock needs to be removed by the ice (especially when considering greater erosion underneath thicker, temperate ice in glacial troughs or on trough flanks and potentially shoulders; cf. line 426f.). In some cases, we know that sediment thickness can exceed several deka-metres (e.g. tunnel valleys), in some overdeepened valleys almost as much as on fjord floors, i.e. a kilometre and more (cf. Preusser, F., Reitner, J.M. & Schlüchter, C. Distribution, geometry, age and origin of overdeepened valleys and basins in the Alps and their foreland. *Swiss J Geosci* 103, 407–426 (2010). <https://doi.org/10.1007/s00015-010-0044-y>). While some of this sediment will be preserved throughout multiple glacial cycles and added to under favourable conditions, you currently only deal with this by stating areas near EIS ice divides where we know erosion was less efficient. However, I struggle to see how thick areas of sediment in the lowland areas (and those between ice divides and those lowland areas near the margins) would NOT affect an averaged erosion rate calculated from bedrock alone, i.e. with those sediments that make up the vast majority of the ice sheet bed in ~30-~50% of the area covered by it. Certainly, if those sediments would have been stripped away at least partially, it would make the erosion rates you calculate more conservative, i.e. they could potentially underestimate the true erosive capability of the entire EIS. I have no idea how much of an underestimate this would give, but I would suggest putting this caveat into the main discussion rather than hide it in the methods – it is a central problem! One way to get a quick estimate, if the authors wanted to include a number/bracketing values, might be to use a sediment thickness map (base of the Quaternary) that is available for several countries. I also note that such eroded sediment would be included in all of the TMFs, including the one you use to derive catchment erosion rates. This means that data from this catchment could also fairly easily be included in any calculations for the entire EIS and perhaps give a more reliable estimate of net erosion at ice-sheet scale?

At any rate, I would strongly recommend you include this caveat explicitly rather than leaving it in its implicit form when talking about limited erosion at ice divides. Perhaps you could include a few references to work from the EIS that has shown partial preservation of such sediments in areas further away from the ice divides such as southern and central Sweden, and outside the mountains?

As you point out, this issue of budgeting the erosion of previously eroded sediments is not explicitly incorporated in the calculated erosion rates. Unfortunately this is a problem for most source-sink erosion-sediment budgets. For the sediment volumes at least, there is an assumption in the backstripping of the Younger Basement and Platform provinces that at the beginning of each new glacial the ice sheet will need to evacuate a similar amount of sediment cover. The erosion and delivery of these sediments to the TMF are thus essentially phase-lagged, and should cancel out if we assume the ice sheet leaves behind a similar thickness/volume of till sheets after each glaciation. In other words we indirectly compensate for this process to an extent because the volume of sediments in the TMF packages we are backstacking include sediments that were eroded during MIS6/Saalian but could not reach the TMF until the Weichselian.

However, the trickier problem to constrain is that this initial sediment evacuation will affect the erosion rate calculation by shortening the timeframe available for fresh bedrock erosion, as you say leading to more conservative rate calculations. On land, the areas outside core zones with old sediment tend to be described in terms of depositional age e.g. a till bed of Late Weichselian age. But sediment in that till bed may have been eroded during previous glaciations. Yet, these average Late Weichselian sediment depths are generally small – 6m for southern Sweden, for example, and as mentioned in the point above it is possible that such sediments could be eroded rapidly, at rates up to 1 m a^{-1} under fast-flowing ice.

We take your recommendation to explain this caveat in the main text, and have expanded upon this with a paragraph at the end of the section ‘Empirical erosion scaling constraints’, including citations exemplifying areas of sediment preservation beyond the core regions.

3. Suspended sediment load

I am not quite sure what these data add to the overall argument. They do seem to be an add-on, but one that can be done without. My main argument here is that abrasion clearly produces rock flour that would be transported as suspended load, but this would most likely not make it from the source (point of abrasion) to the margin in most cases, but be deposited from suspension along the flowline of the ice sheet (e.g. in subglacial lakes or mixed in with other sediments such as subglacial traction till). By only including suspended load, you also ignore (at least explicitly as this is not stated in the text) the contribution of abrasion versus plucking in bedrock erosion. In any case, I would recommend adding a sentence or two to the methods and discussion. At present, the rationale for including suspended sediment load escapes me, unfortunately.

We feel this is an important aspect to the paper as it provides a link between our palaeo insights to observations made from contemporary glacial systems. The technique of using suspended sediment fluxes as a proxy for estimating contemporary erosion rates is not uncommon (e.g. Cowton et al., 2012). Our unique time-transgressive framework thus provides an opportunity to demonstrate how such estimations are likely very transient (geologically speaking), poor context for the longer-term erosional history of the ice sheet, and that such inferred erosion rates do not necessarily reflect current ice-bed interactions (ie. the phase-lagged nature of these mobilized sediments when the subglacial system is flushed during periods of high surface melt). We preface this section with some further sentences to better demonstrate this rationale.

Figures

These are of a very high quality throughout, and my comments are fairly minor, but hopefully they help clarify the visual message.

Fig. 2: “Celtic” (ice sheet): Sorry to say, but this is a historically-nonsensical misnomer: The Celts actually inhabited much of continental Europe all the way to S Spain and into Eastern Europe way beyond any pre-Weichselian ice limits! The accepted term for this sector of the EIS is the British-Irish Ice Sheet (BIIS). You also mention “British” somewhere in the text.

Reference to the Celtic ice sheet has been removed from the figure. Reference to the ‘British Isles’ in the main text has also been modified to reduce ambiguities over geographical specificity.

The figure needs a legend that decodes the TMFs. A legend has been added

Why are the TMFs in the NW sector of the BS sector delineated by unnatural-looking straight lines? They have presumably been mapped (i.e. their exact planforms could be represented) like the others? The morphology of the TMFs in this sector are poorly constrained due to the relative sparsity of echosounder data (Batchelor and Dowdeswell, 2014; Jakobsson et al., 2014; Streuff et al., 2022), and we are not aware of any other publications that have them mapped in detail. The planforms used here are from Batchelor & Dowdeswell, with a citation now added to the caption.

Could you please include a stippled line to show the maximum extent of the ice sheet during the last glaciation, so that the areas you present erosion envelopes for can be put into its full geographical context, in particular to the ice margins? Empirical ice margins from the three major stadials are now shown.

Fig. 3: A: Include the abbreviations from the caption in a legend. Also decode the white dots in the same legend. The location of panel B is missing from A (include a rectangle or dot) - it took me a while to locate B in A. Legend items have been added, and the mapped sediments in B outlined.

B: I like this figure! One plea: wouldn't it be better and more intuitive to convert the thickness measurement from metres per second into true thickness in metres? I am suggesting this, because that way the figure could also be more easily included for both research and teaching purposes. Thanks. It seems this was our intention given the description in the caption. 3B has been updated accordingly

C: The scales to me seem odd: I would expect “(…)” for the units rather than a “/” Changed.

Fig. 4: “Point density cloud” rather than “map” would be more intuitive to me. Changed

Table 1: Is there a reason “Y.B.” has dots as the only abbreviation in the ms? The acronym is now removed

“Y.B. plateau (≥ 550 m a.s.l.)”: Not all sites above 550 m qualify as plateaux, neither in Scandinavia, Scotland nor elsewhere. This to me is a gross oversimplification that should either be clarified or removed altogether. The most un-ambivalent and uncomplicated way of doing this would be to replace this with “Y.B. high elevation (≥ 550 m a.s.l.)”. The description has been changed as suggested.

Fig. S3: A: the unit in the top right-hand corner is unclear to me (typo?). Do you mean volume or area here? Units are corrected and the colour ramp extended to the whole data range.

General observations

- There are a few minor, but still irritating problems with the correct hyphenation of compound adjective-noun adjectives versus normal adjective and noun constructions (for example, lines 33-34: it should be “low-relief continental shields”, but never “high-latitudes”). We have gone through to find any inconsistencies in hyphenation.

- 35: “the drivers of and controls on” changed

- 98 (and elsewhere): “GIII” – this is not specified anywhere else in either text or figures. Please decode and be more specific. A citation is now given for this regional stratigraphic framework. The GIII unit is already referred to in Fig 3B, Figure S2 and Table S2

- 108: “T formation” – this is not specified anywhere else in either text or figures. Please decode and be more specific. Additional text has been added describing the Naust Formation

- 112: “areas” – should be “volumes”? areas, but units were wrong.

- 147: “we do find evidence”. This is very interesting! What evidence is this specifically? And what does it indicate? Please provide more detail here on the hierarchy of controls and type of evidence you have (is it ‘hidden’ in one of the figures and could be highlighted, for example? Does it come out of particular model runs?). Otherwise, if you’re unable to, I suggest you drop this general statement, because it is a bit of a mean teaser. Changed to ‘our results are consistent with...’. Each of these specific controls is discussed more thoroughly in the subsequent paragraphs

- 157: “perma-frozen clays”: Is it really just clays or better to state “sediments”? changed to ‘clayey sediments’. According to Astakhov they are clay rich – the Meso-Cenozoic sedimentary source rocks are 35-40% marine clay.

- 161: should be “primary controls” changed

- 251: decode GrIS in previous line acronym removed

- 254: would “insights into the transient nature of these processes” be clearer? I struggle with the idea of an insight being transient (i.e. fleeting) agreed

- References: There are a few incomplete bibliographic entries: numbers 39, 57, 75 and 79 need attention. These references have been adjusted.

I hope these comments are helpful in rounding the message of this highly interesting and topical ms even further.

Please do not hesitate to get in touch directly with me as well if you want to discuss any of my comments.

Sven Lukas, LU

Reviewer #3:

Review of " The extreme yet transient nature of subglacial erosion" (NCOMMS-22-06881)

Reviewer: Simon Cook, University of Dundee, UK

Summary

I was grateful for the opportunity to review this manuscript by Patton et al. Determining the controls on glacial erosion rates is an important and interesting subject, with many fundamental issues still unresolved despite many decades of effort. The authors present a wide-ranging and hard-won set of results, including marine geophysical, geological and geochronological data, and outputs from the modelling of ice sheet dynamics over the last glacial cycle. The sheer scale of the Eurasian Ice Sheet, spanning a huge latitudinal range, allows the authors to test some key hypotheses about what controls glacial erosion beneath ice sheets. They arrive at the conclusion that ice sheet thermodynamics is the dominant factor that determines glacial erosion rates, and that other factors that have previously been held up as the main controls (climate/latitude, lithology, etc.) are less significant. I'm broadly supportive of this effort and its conclusions, and I think the sheer spatial and temporal scale under consideration here, as well as the depth of analysis make it very impactful, and hence a good fit with Nature Communications. I have some broader comments followed by minor issues. I hope the authors find these helpful in refining their work, though it is already very well written and seems to have been put together with great care.

- Article title. One to consider, but not a deal-breaker by any means... I wonder if the title of the article is truly reflective of the key message of the paper. I like the title, and I can see the appeal of using words like 'extreme', but it strikes me that this title is more reflective of the observations of the paper rather than its actual conclusions (i.e. that ice dynamics regulates erosion rates). I think people are already aware that erosion can be variable; they may not be so aware of the importance of ice thermodynamics, which is perhaps where the real novelty is here. Finally, the maximum erosion rate found in this study (5.2 mm/year) is not the extreme limit of glacial erosion – Alaskan tidewater glaciers have been found to erode at 10s of mm per year.

Our case for the use of extreme was based around the perception of polar ice sheets on average erode at several orders of magnitude slower than indicated by our results.

- The role of latitude, climate and "latitude/climate" in glacial erosion. Getting straight to the point, I don't think what I discuss below will need much to fix, but I think it is important that the authors are clear about what they are testing when it comes to the latitudinal/climatic control on glacial erosion because it is one of the central hypotheses that they test in this study. The authors tend to equate latitude with climate ("latitude/climate"), which is a very crude equivalence. Ultimately, I think the authors have tested the relationship between latitude and glacial erosion rate, and not necessarily climate (by which I mean temperature and precipitation) and erosion rate.

As we demonstrated in Cook et al. (2020), latitude is probably only a very crude proxy for climate, and even then only for temperature, and not precipitation. In our dataset, we lumped all sorts of glaciers together and found only a very weak relationship between latitude and climate. Koppes et al. (2015) did find a relationship, but only because they isolated latitude as a controlling variable by sampling only tidewater glaciers across a wide latitudinal range. So I think the results of Cook et al. (2020) support your results of only weak correlation between latitude and erosion rate.

Around L144-149, a direct comparison is made between the results of this study and the results of previous studies. That's important to do of course, but I think there is a bit of missing nuance here. Previous work (Koppes et al., 2015, Anderson et al., 2016, and Cook et al., 2020 are cited in the preceding sentence of the manuscript) has examined the relationship between latitude, and to a lesser extent, climate, for valley glaciers. Actually, to be fair, Cook et al. mixed in a few ice sheet outlet glaciers, which on reflection is perhaps a bit unhelpful in this context. Nonetheless, should we really expect ice sheet erosion to be limited by the same climatic/latitudinal constraints as valley glaciers? It's hard to tell exactly from the images here (nice as they are), but it looks like some of the glaciers draining the EIS crossed very large latitudinal ranges themselves – possibly crossing geological boundaries, possibly passing through different climates (temperature and precipitation regimes), perhaps snow falling in one place and feeding an ice stream that does most of its erosion hundreds of km away, and so on. My point being that this is on a totally different scale and context to the sorts of glaciers that have been used in the past to determine sliding/climate/latitude-erosion relationships. Maybe latitude works better for carefully selected valley glaciers (Koppes et al., 2015; Anderson et al., 2016) than it does for ice sheets (this study) or less-choosy selections of glaciers (Cook et al., 2020). I just wonder whether the authors need to be a little less forthright in their conclusions and acknowledge these nuances (see for example L266).

Even if the authors consider that they are examining the relationship between climate and erosion rate, I still wonder if what they really mean is temperature, or latitudinally controlled temperature. And as we showed in Cook et al. (2020) for (mostly) valley glaciers, precipitation is also very important. But I take the authors' overall point that ice sheets are far more able to set their own rhythm of glacial erosion.

We appreciate the reviewer's comments on this point, and agree that these nuances and differences should be incorporated better. As such we have reworded the initial paragraph of the discussion, including how the differences in glacier scale could explain why results diverge from these cited works. This adjustment sets up nicely our hypothesis in the last sentence for ice sheets modulating their own pace of erosion over other environmental controls.

A sentence in the conclusion is also modified to reflect this nuance.

Minor points, typos, etc.

L36-39. Since you're giving examples, I think you need some references here. The citations from the previous sentence are placed here

L49. What exactly is meant by "kilometre-scale denudation"? Removed the word processes. Essentially, total erosion through the Quaternary is at the kilometre scale.

L100. Maybe I've missed this somewhere, but how do you split the total volume of the GIII unit (62,105km³) into three glacial advances and end up with a value of 37,263km³ for the most recent glacial cycle? Glacial cycle duration? Some geophysical and geochronological evidence? There are five seismic sequences in GIII reflecting distinct phases of deposition. Seismic data resolution is not good enough to partition volume estimates further, so we simply assume that the 3 Weichselian advances account for 3/5ths of the total Weichselian and Saalian volume.

L108 – what is the "T formation"? I know there are citations, but I think it needs to be explained in this paper. Added some explanation here.

L112 – areas of erosion with units of km³. Should these be km²? changed

L112 – "respectively" is used, but it's not clear in what order the TMF/catchments are being

considered from the text here. Edited to qualify better

L117 – what is the “LR04 stack”? First time this has been mentioned and it’s not clear what we’re dealing with. I’m guessing some geophysically-derived sediment facies from one of these TMFs? A brief description is now given.

L126, though a general point. Here, it is stated that the total eroded volume has been 126,518 km³. That seems super-precise! Elsewhere, you’ve quoted areas to the nearest thousand, and then other volumes (L99 and L102) to the nearest km³. Given the uncertainties in geophysical measurements, dating, ice dynamics and so on, I’m wondering whether these very precise numbers are really warranted. We have rounded the reported volume to the nearest 100 km³ here in the main text.

L135. I’m not sure you should use the term ‘British Isles’ given that the Celtic Ice Sheet covered the island of Ireland too. The ‘British Isles’ does encompass the island of Ireland technically, but I agree it’s not intuitive terminology. Changed to Great Britain, and Ireland.

L143. “correlate relationships” seems an odd phrase. Changed to ‘link’

L143. Not sure the tone is quite right here. “...climate or latitudinal proxies alone have met with limited success”. I’d argue there has been progress rather than limited success. Sentence reworded to emphasise the positive progress

L165. This diagram (Fig 3C) doesn’t show the result alluded to in the text here. I disagree - in this figure observations from both Platform and Younger Basement provinces overlap, and the erosion laws scaling the mean erosion rate (ie total erosion normalized for glacial occupancy) with ice discharge are relatively similar, certainly when both are compared to the Shield province.

L176-177. This point about negative feedback loops could be better explained. Rather vague. I guess what you mean is that retrograde slopes induce glaciohydraulic supercooling, which leads to low sediment transport capacity in subglacial streams, hence deposition of a protective till layer that limits erosion. If that’s the case, then I guess you don’t want to use so many words to say all that, but I really think you should at least cite Alley et al. (2003) Nature (Stabilizing feedbacks in glacier bed erosion, I think). But what I’m not clear on here is that this passage of text is referring to linear erosion of fjords, whereas what you are discussing in this final sentence is about overdeepenings. I know fjords commonly have overdeepenings, but you don’t actually say that, so it seems like a logic step is missing here. Citation to Alley 2003 has been added, and these sentences reworded to make the linkage with supercooling across potential retrograde slopes in fjords clearer.

L214. Shouldn’t this refer to Fig S5? Yes, changed

Fig 1. Does the y-axis label for SSL need to be coloured purple to match the line colour? I realise this might not be straightforward to do in the code, and possibly more hassle than it’s worth. I made a workaround to do this now

L225 and whole section on suspended sediments. Can we be clear here: what do you mean by suspended sediment loads (SSL)? Is this a catch-all term for all sediment exported in subglacial and proglacial streams, including bedload? Or is it only suspended sediment, in which case these numbers may only represent minimum sediment export estimates because they ignore bedload? As you’re probably aware, bedload transfer is rarely measured in sub-/pro-glacial systems; most glacial erosion estimates are based on suspended sediment loads measured at a meltwater portal. But this ignores bedload, or only considers it using some fudge factor. I’m curious to know what is considered in this study, or what the assumptions are. As the suspended-sediment flux estimates follow closely the analyses by Overeem et al. (2017) using satellite imagery, these do not include any bedload components. We have expanded this section in the Methods to describe in more detail how these estimates were derived and on the reported relative errors.

L254-6. I think this is an important point about the extrapolation of modern glacial erosion rates and sediment fluxes to longer-term studies. Others have made this same point, including Koppes et al. (2015), Cook et al. (2020), Jaeger and Koppes (2016), and so on. Perhaps a citation or two could be

added here. Citations have been added

L290. "Version 1.4" currently in superscript, but probably should be placed in brackets in regular font, before the citation number. Changed

L318 and L321. Inconsistent font style for term, "a". fixed

L325. Suggest reword to, "For simplicity, the values of...to be constant⁵⁸" changed

L333. "Ice experiments" seems an odd/vague term. Changed to ice-modelling experiments

L699. data are fixed

Fig 3. Do you need r^2 and p-values for your lines of best fit? No, the erosion scaling laws are not lines of best-fit and are independent from these plotted empirical observations.

Andersen, K.K., Azuma, N., Barnola, J.-M., Bigler, M., Biscaye, P., Caillon, N., Chappellaz, J., Clausen, H.B., Dahl-Jensen, D., Fischer, H., Flückiger, J., Fritzsche, D., Fujii, Y., Goto-Azuma, K., Grønvold, K., Gundestrup, N.S., Hansson, M., Huber, C., Hvidberg, C.S., Johnsen, S.J., Jonsell, U., Jouzel, J., Kipfstuhl, S., Landais, A., Leuenberger, M., Lorrain, R., Masson-Delmotte, V., Miller, H., Motoyama, H., Narita, H., Popp, T., Rasmussen, S.O., Raynaud, D., Rothlisberger, R., Ruth, U., Samyn, D., Schwander, J., Shoji, H., Siggard-Andersen, M.-L., Steffensen, J.P., Stocker, T., Sveinbjörnsdóttir, a E., Svensson, A., Takata, M., Tison, J.-L., Thorsteinsson, T., Watanabe, O., Wilhelms, F., White, J.W.C., 2004. High-resolution record of Northern Hemisphere climate extending into the last interglacial period. *Nature* 431, 147–151. <https://doi.org/10.1038/nature02805>

Batchelor, C.L., Dowdeswell, J.A., 2014. The physiography of High Arctic cross-shelf troughs. *Quaternary Science Reviews* 92, 68–96. <https://doi.org/10.1016/j.quascirev.2013.05.025>

Cowton, T., Nienow, P., Bartholomew, I., Sole, A., Mair, D., 2012. Rapid erosion beneath the Greenland ice sheet. *Geology* 40, 343–346. <https://doi.org/10.1130/G32687.1>

Hall, A., van Boeckel, M., 2020. Origin of the Baltic Sea basin by Pleistocene glacial erosion. *GFF* 142, 237–252. <https://doi.org/10.1080/11035897.2020.1781246>

Jakobsson, M., Andreassen, K., Bjarnadóttir, L.R., Dove, D., Dowdeswell, J.A., England, J.H., Funder, S., Hogan, K., Ingólfsson, Ó., Jennings, A., Krog Larsen, N., Kirchner, N., Landvik, J.Y., Mayer, L., Mikkelsen, N., Möller, P., Niessen, F., Nilsson, J., O’Regan, M., Polyak, L., Nørgaard-Pedersen, N., Stein, R., 2014. Arctic Ocean glacial history. *Quaternary Science Reviews* 92, 40–67. <https://doi.org/10.1016/j.quascirev.2013.07.033>

Overeem, I., Hudson, B.D., Syvitski, J.P.M., Mikkelsen, A.B., Hasholt, B., van den Broeke, M.R., Noël, B.P.Y., Morlighem, M., 2017. Substantial export of suspended sediment to the global oceans from glacial erosion in Greenland. *Nature Geoscience* 10, 859–863. <https://doi.org/10.1038/ngeo3046>

Patton, H., Hubbard, A., Andreassen, K., Auriac, A., Whitehouse, P., Stroeven, A.P., Shackleton, C., Winsborrow, M.C.M., Heyman, J., Hall, A.M., 2017. Deglaciation of the Eurasian ice sheet complex. *Quaternary Science Reviews* 169, 148–172. <https://doi.org/10.1016/j.quascirev.2017.05.019>

Patton, H., Hubbard, A., Andreassen, K., Winsborrow, M., Stroeven, A.P., 2016. The build-up, configuration, and dynamical sensitivity of the Eurasian ice-sheet complex to Late Weichselian climatic and oceanic forcing. *Quaternary Science Reviews* 153, 97–121. <https://doi.org/10.1016/j.quascirev.2016.10.009>

Streuff, K.T., Ó Cofaigh, C., Wintersteller, P., 2022. GlaciDat – a GIS database of submarine glacial landforms and sediments in the Arctic. *Boreas* 51, 517–531. <https://doi.org/10.1111/bor.12577>